# Revisiting the Bertrand Paradox via Equilibrium Analysis of No-regret Learners

**Arnab Maiti** [1]  **Junyan Liu** [1]  **Kevin Jamieson** [1]  **Lillian J. Ratliff** [1]

## Abstract

We study the discrete Bertrand pricing game with a non-increasing demand function. The game has $n \geq 2$ players who simultaneously choose prices from the set $\{1/k, 2/k, \ldots, 1\}$, where $k \in \mathbb{N}$. The player who sets the lowest price captures the entire demand; if multiple players tie for the lowest price, they split the demand equally.

We study the Bertrand paradox, where classical theory predicts low prices, yet real markets often sustain high prices. To understand this gap, we analyze a repeated-game model in which firms set prices using no-regret learners. Our goal is to characterize the equilibrium outcomes that can arise under different no-regret learning guarantees. We are particularly interested in questions such as whether no-external-regret learners can converge to undesirable high-price outcomes, and how stronger guarantees such as no-swap regret shape the emergence of competitive low-price behavior. We address these and related questions through a theoretical analysis, complemented by experiments that support the theory and reveal surprising phenomena for no-swap regret learners.

## 1. Introduction

The concept of price competition in a duopoly dates back to Bertrand (1883). In his critique of Cournot's work, Bertrand introduced the idea that firms choose prices for their products and sell in order to maximize profits. He argued that consumers purchase from the firm charging the lower price, an observation later elaborated and extended by Edgeworth (1897). This framework of competition through prices is commonly referred to as the Bertrand pricing game. Formally, in a Bertrand pricing game, each player chooses a price, and the player who sets the lowest price captures the entire demand at that price, which we refer to as the

[1]University of Washington. Correspondence to: Arnab Maiti <arnabm2@uw.edu>.

*Proceedings of the 43rd International Conference on Machine Learning*, Seoul, South Korea. PMLR 306, 2026. Copyright 2026 by the author(s).

transaction price. If multiple players tie for the lowest price, demand is split equally among them. A player's utility is the product of its per-unit margin (the price it sets minus marginal cost) and the demand it serves.

Under symmetric marginal costs, where every player has the same marginal cost, any firm can profitably undercut the lowest posted price by an arbitrarily small amount and capture the entire demand, thereby obtaining higher utility. As a consequence, high prices cannot be sustained in the Bertrand pricing game, and prices are driven close to marginal cost. However, in real markets this is clearly not the case, as firms often set prices well above marginal cost (Tremblay & Tremblay, 2019; Dufwenberg & Gneezy, 2000; Hall, 1988). This phenomenon is famously known as the Bertrand paradox, and it has spurred a large body of work aimed at resolving it (Geromichalos, 2014; Brander & Spencer, 2015; Cabon-Dhersin & Drouhin, 2010; Baye & Morgan, 1999; Hehenkamp, 2002; Carvalho, 2009).

One approach to resolving the paradox is to model the setting as a repeated game, where players set prices over multiple rounds after observing prices chosen in previous rounds, rather than treating it as a one-shot interaction. This line of work has received substantial attention in recent years (Nadav & Piliouras, 2010; Bruttel, 2009; Arunachaleswaran et al., 2024; Bichler et al., 2024; Farrell, 2000). When players choose prices repeatedly with the objective of maximizing long-run profits, a central concept studied in game theory is regret, which measures the increase in utility a player could obtain by unilaterally deviating from its realized sequence of prices. Two classes of deviations are commonly considered: deviations to a fixed price and deviations that swap one chosen price with another. Regret with respect to the former is known as external regret, while regret with respect to the latter is known as swap regret (Cesa-Bianchi & Lugosi, 2006). It is well known that when all players incur low external regret, the empirical joint distribution of prices converges to a coarse correlated equilibrium (CCE) (Roughgarden, 2016). Likewise, when all players incur low swap regret, the empirical joint distribution converges to a correlated equilibrium (CE) (Roughgarden, 2016). These convergence results motivate the study of such equilibria attainable by no-regret learners in the Bertrand pricing game as a means of understanding when high transaction prices can be sustained and when competitive forces drive the

transaction prices downward.

While correlated equilibrium has been extensively studied (Wu, 2008; Jann & Schottmüller, 2015; Arunachaleswaran et al., 2024), coarse correlated equilibrium remains comparatively less understood. Only recently, Nadav & Piliouras (2010) showed that under symmetric costs there exists a demand function for which a CCE yields high utilities, suggesting that no-external-regret learners can sustain high transaction prices. It remains open whether such a guarantee extends to arbitrary non-increasing demand functions, and how the resulting transaction prices scale with the number of firms $n$. Related questions also arise in asymmetric-cost duopolies, where one firm has a significantly higher marginal cost than the other. Similarly, it is unclear whether a no-swap-regret learner can always drive the transaction prices down when competing against a no-external-regret learner.

While a subset of recent work has focused on the continuous price setting (Nadav & Piliouras, 2010; Jann & Schottmüller, 2015; Raskovich, 2025), our focus is on a discrete price setting, motivated by the fact that real-world markets operate with discrete prices (for example, the U.S. dollar is discretized into cents) and by recent work adopting this perspective (Calvano et al., 2020; Eschenbaum et al., 2022; Arunachaleswaran et al., 2024; Collina et al., 2025). These considerations motivate the following central question:

> *What transaction prices can be sustained in the equilibrium outcomes attainable by no-regret learners in a discrete Bertrand pricing game with an arbitrary non-increasing demand function?*

### 1.1. Problem Setting

Toward answering the question posed above, we first formally define the Bertrand pricing game. The game consists of $n$ players, and each player simultaneously chooses a price from the discrete set $\mathcal{P} := \{i/k : i \in [k]\}$, where $k$ is a positive integer. Let $f : \mathcal{P} \to [0, 1]$ be a non-increasing demand function, i.e., $f(x) \geq f(y)$ for all $x \leq y$. Let $c_i \in \mathcal{P} \cup \{0\}$ denote the marginal cost of player $i$. Given a price profile $(x_1, \ldots, x_n) \in \mathcal{P}^n$, the utility of player $i$ is defined by

$$u_i(x_i, x_{-i}) = \begin{cases} \dfrac{(x_i - c_i), f(x_i)}{|j \in [n] : x_j = x_i|} & \text{if } x_i = \min_{j \in [n]} x_j, \\ 0 & \text{otherwise,} \end{cases}$$

where $x_{-i}$ denotes the prices chosen by all players other than $i$. For any price tuple $x \in \mathcal{P}^n$, we define the transaction price as $\min_{j \in [n]} x_j$. We also define the maximum monopoly utility of the player $i$ as $\max_{x \in \mathcal{P}}(x - c_i)f(x)$.

In this paper, we analyze equilibrium notions in the Bertrand pricing game. To this end, we first define $\Phi$-correlated

equilibrium. For each player $i \in [n]$, let $\Phi_i$ denote a collection of deviation maps $\phi_i : \mathcal{P} \to \mathcal{P}$, and let $\Phi := (\Phi_1, \Phi_2, \ldots, \Phi_n)$. A joint distribution $\mathcal{D}$ over price tuples $x = (x_1, x_2, \ldots, x_n) \in \mathcal{P}^n$ is a $\Phi$-*correlated equilibrium* if, for every player $i \in [n]$ and every deviation map $\phi_i \in \Phi_i$, we have

$$\mathbb{E}_{x \sim \mathcal{D}}[u_i(x)] \geq \mathbb{E}_{x \sim \mathcal{D}}\big[u_i\big(\phi_i(x_i), x_{-i}\big)\big].$$

If, for each player $i$, the set $\Phi_i$ consists of all constant deviation maps, i.e.,

$$\Phi_i := \{\, \phi_p : \mathcal{P} \to \mathcal{P} \mid p \in \mathcal{P}, \ \phi_p(x) = p \ \forall x \in \mathcal{P} \,\},$$

then $\Phi$-correlated equilibrium coincides with coarse correlated equilibrium (CCE). In contrast, if for each player $i$ the set $\Phi_i$ contains all possible deviation maps, i.e.,

$$\Phi_i := \{\, \phi : \mathcal{P} \to \mathcal{P} \,\}.$$

then $\Phi$-correlated equilibrium coincides with correlated equilibrium (CE).

As we study the equilibrium notions above, it is also important to define the corresponding regret notions whose minimization leads to convergence to these equilibria. Given a sequence of price profiles $x^{(1)}, x^{(2)}, \ldots, x^{(T)}$, where each $x^{(t)} = (x_1^{(t)}, x_2^{(t)}, \ldots, x_n^{(t)}) \in \mathcal{P}^n$, the $\Phi_i$-regret of player $i \in [n]$ is defined as

$$R_{\Phi_i}(T) := \max_{\phi_i \in \Phi_i} \sum_{t=1}^{T} u_i\big(\phi_i(x_i^{(t)}), x_{-i}^{(t)}\big) - \sum_{t=1}^{T} u_i\big(x_i^{(t)}, x_{-i}^{(t)}\big).$$

A learning algorithm for player $i$ is called a *no-$\Phi_i$-regret* learner if $R_{\Phi_i}(T)/T \to 0$ as $T \to \infty$. If $\Phi_i$ consists only of constant deviation maps, then $R_{\Phi_i}(T)$ coincides with external regret and the corresponding learner is called a *no-external-regret* learner. If instead $\Phi_i$ contains all possible deviation maps, then $R_{\Phi_i}(T)$ coincides with swap regret and the corresponding learner is called a *no-swap-regret* learner.

Moreover, in the duopoly setting where $n = 2$, for every $\Phi$-correlated equilibrium there exist *no-$\Phi_i$-regret* learners for each player $i$ such that, when both players simultaneously use their corresponding learners, the empirical distribution of $\{x^{(t)}\}_{t=1}^{T}$ converges to that equilibrium as $T \to \infty$. We refer the reader to Appendix C.1 for more details.

In this paper, we use the terms *high* and *low* transaction prices informally. In the symmetric-cost setting (all firms have marginal cost $c$), a transaction price $p$ is *high* if its markup $p - c$ is within a constant factor (independent of $k$) of the monopoly-optimal markup $p_* - c$, where $p_*$ maximizes monopoly utility, and *low* otherwise. In the asymmetric duopoly setting ($n = 2$ and $c_1 < c_2$), $p$ is *high* if $p - c_2$

is within a constant factor (independent of $k$) of firm 2's monopoly-optimal markup, and *low* otherwise. We say two no-regret learners *sustain high transaction prices* if transaction prices are high for a constant fraction (independent of $k$) of the interaction rounds; otherwise, we say they *drive prices down*.

## 1.2. Our contributions

Having described the problem setting, we now turn to answering the question posed earlier. We begin by outlining our contributions in the symmetric-cost setting, where all players share the same marginal cost $c$. We show in Theorem 2.1 that, for any non-increasing demand function, there exist two no-external-regret learners that can sustain high transaction prices while competing against each other. In contrast, we show in Theorem 2.2 that, for any non-increasing demand function, any pair of no-swap-regret learners drive the transaction prices down when competing against each other. We then ask whether a no-swap-regret learner can always drive the transaction prices down when competing against a no-external-regret learner. We answer this question in the negative in Theorem 2.3 by showing that there exists a demand function and a pair of learners, one no-external-regret and one no-swap-regret, such that they sustain high transaction prices while competing against each other. Moving beyond duopoly, we show in Theorem 2.4 that for $n \geq 3$ and any non-increasing demand function, the presence of just two no-swap-regret learners suffices to drive the transaction prices down. Finally, we show in Theorem 2.5 that as the number of firms grows, the highest utilities that no-external-regret learners can achieve decay exponentially when they compete against each other.

We next extend our results for the Bertrand duopoly to the asymmetric-cost setting, where player 1 has a lower marginal cost than player 2, and and the gaps $c_2 - c_1$ and $1 - c_2$ are fixed positive constants, independent of $k$ and much larger than $1/k$. First, in Theorem 2.6 we show that, for a broad class of demand functions, there exist two no-external-regret learners that can sustain high transaction prices while competing against each other. We next show in Theorem 2.7 that, for any non-increasing demand function, any pair of no-swap-regret learners drive player 2's utility to zero when competing against each other. Finally, in Theorem 2.8 we show that there exist marginal costs, a demand function, and a $\Phi$-correlated equilibrium in which both players obtain expected utility that is a constant fraction of the maximum utility achievable under monopoly, even when $\Phi_1$ consists only of constant deviation maps while $\Phi_2$ consists of all deviation maps. The same guarantee also holds under the roles reversed, i.e., when $\Phi_1$ contains all deviation maps and $\Phi_2$ contains only constant deviation maps.

In Section 3, we first run numerical experiments that compute, for several well-known demand functions, the exact fraction of the maximum monopoly utility achieved by the best possible CCE both in the symmetric and asymmetric-cost setting. Interestingly, in the symmetric-cost setting, we observe that this fraction converges to roughly $1/e$. We then study, again under symmetric costs, how the maximum achievable sum of players' expected utilities over all CCE varies with the number of firms $n$. Across several standard demand functions, we observe an exponential decay as $n$ increases, with utility becoming negligible beyond five firms. Finally, in the asymmetric-cost duopoly setting with $c_1 < c_2$ and constant demand, we run experiments with standard no-swap-regret learners, illustrating how different correlated equilibria can emerge from competition between such learners. A surprising trend is that the transaction prices can fall well below $c_2$, and this outcome can vary with algorithmic choices such as the learning rate.

## 1.3. Related Works

Prior work has explored several approaches for resolving the Bertrand paradox, including capacity constraints (Peters, 1984; Geromichalos, 2014), product differentiation (Anderson, 2008; Brander & Spencer, 2015), consumer search (Stahl, 1989), imperfectly informed customers (Hehenkamp, 2002), repeated interactions (Nadav & Piliouras, 2010; Bruttel, 2009; Arunachaleswaran et al., 2024; Bichler et al., 2024; Farrell, 2000), and collusion (Spagnolo, 2000; Melkonyan et al., 2017; Obara & Zincenko, 2011).

Among recent works on repeated interactions, Arunachaleswaran et al. (2024) is particularly closely related to ours. They study the discrete Bertrand pricing game with constant demand $f(x) = 1$ and zero marginal costs. Their main result shows that, in a duopoly, if player 1 is a no-swap regret learner, there exists an algorithm for player 2 that yields high utility for both players. They also show that, under any correlated equilibrium, each player's expected utility is at most $O(1/k)$. They also study mean-based algorithms and showed that these algorithms drive the prices down.

Collina et al. (2025) also consider the discrete Bertrand pricing game with constant demand and zero marginal costs. Building on techniques of Feldman et al. (2016), who established a similar phenomenon for auctions, they observe that players' maximum attainable utilities decay exponentially with the number of players. Jann & Schottmüller (2015) characterize correlated equilibria of the Bertrand pricing game in the continuous-price setting for arbitrary non-increasing demand functions. Nadav & Piliouras (2010) likewise study the continuous-price model and show the existence of a high-utility CCE for linear demand. In the discrete setting, Bichler et al. (2024) use numerical experi-

ments to demonstrate that high utility can be achieved via CCE for linear demand, and Wu (2008) characterize correlated equilibria for linear demand.

Beyond Bertrand pricing games, CE and CCE have also been studied in other oligopoly models. Liu (1996) and Yi (1997) study correlated equilibrium in Cournot oligopoly. Ray & Gupta (2013) study CCE in linear duopoly games, and Moulin et al. (2014) study CCE for a class of symmetric two-player quadratic games. Awaya & Krishna (2020) study CCE in the context of cartels. More broadly, CE has been studied in potential and concave games, which include a wide range of settings encompassing oligopoly models (Neyman, 1997; Ui, 2008). Finally, Einy et al. (2022) introduce a notion of strong robustness and connect it to CE in Cournot and Bertrand oligopoly.

## 2. Results

In this section, we formally state our main results. We begin by considering the symmetric-cost setting, where all players have the same marginal cost, in Section 2.1. We then extend our analysis to the asymmetric-cost setting for the Bertrand duopoly, where players have different marginal costs, in Section 2.2. The omitted detailed proofs are presented in Appendix A.

### 2.1. Symmetric costs setting

We begin with the Bertrand duopoly and assume that both players have the same marginal cost $c$. Prior work, including Nadav & Piliouras (2010) and Bichler et al. (2024), has observed that for a linear demand function there exists a CCE under which the players obtain utilities that are a constant fraction of the maximum monopoly utility. This raises an important question: for an arbitrary non-increasing demand function, does there exist a CCE under which the players achieve utilities that are a constant fraction of the maximum monopoly utility? We answer this question in the affirmative in the following theorem.

**Theorem 2.1.** *Consider any non-increasing demand function* $f : \mathcal{P} \to [0, 1]$ *and the symmetric-cost setting, where all players have the same marginal cost $c$. There exists a CCE $\mathcal{D}$ such that, for each player $i \in \{1, 2\}$, the following holds:*

$$\mathbb{E}_{x \sim \mathcal{D}}[u_i(x)] \geq \frac{1}{4e^2} \cdot \max_{x \in \mathcal{P}}(x - c)\, f(x).$$

*In other words, for any non-increasing demand function, there exist two no-external-regret learners that can sustain high transaction prices while competing against each other.*

*Proof Sketch.* We now describe a way to construct a symmetric CCE where both players always choose the same price. Let $s_i := (i/k - c) \cdot f(i/k)$ and $S_i := \max_{j \leq i} s_j$.

Let $s_{\max} := \max_{i \in [k]} s_i$ and consider the smallest index $m$ such that $m \in \arg\max_{i \in [k]} s_i$. Consider a constant $\lambda := \frac{1}{2e^2}$ and set $B := \lambda s_{\max}$. We now consider the following two cases.

**Case 1:** $s_{k \cdot c + 1} \geq B$. We choose the price $c + 1/k$ for both the players with probability $1$. Note that this is actually a Nash equilibrium as no player can unilaterally deviate and get a higher reward. Now observe that each player receives a utility of $\frac{1}{2} s_{k \cdot c + 1} \geq B/2 = (\lambda/2) \cdot s_{\max}$.

**Case 2:** $s_{k \cdot c + 1} < B$. Let $i_0 := \min\{i : S_i \geq B\}$. We randomly choose a price $x = i/k$ for both the players with probability $\tau_i - \tau_{i+1}$, where $\tau_i$ is defined as follows:

$$\tau_i := \Pr[x \geq i/k] = \begin{cases} 1, & i < i_0, \\ B/S_i, & i_0 \leq i \leq m, \\ 0, & i > m, \end{cases}$$

Now we prove that we have indeed constructed a symmetric CCE. Due to symmetry, showing that $\mathbb{E}[u_1(i/k, x)] \leq \mathbb{E}[u_1(x, x)]$ for any $i \in [k]$ suffices, where $x$ is the price randomly chosen for both the players as per the distribution above. First, we have the following:

$$\mathbb{E}[u_1(i/k, x)] \leq s_i \tau_i \leq B.$$

We get the last inequality due to the fact that if $i < i_0$, then $s_i < B$ and if $i \geq i_0$ then $s_i \tau_i \leq S_i \cdot \frac{B}{S_i} = B$.

Now, we have the following:

$$\begin{aligned} \mathbb{E}[u_1(x, x)] &\geq \frac{1}{2} \sum_{i=i_0}^{m-1} s_i \cdot \left( \frac{B}{S_i} - \frac{B}{S_{i+1}} \right) + \frac{1}{2} s_m \cdot \frac{B}{S_m} \\ &\geq \frac{B}{2} \sum_{i=i_0}^{m-1} s_i \cdot \left( \frac{1}{S_i} - \frac{1}{S_{i+1}} \right) + B/2 \\ &\qquad\qquad\qquad\qquad\qquad (\text{as } S_m = s_m) \\ &\geq B \end{aligned}$$

The last inequality follows from a sequence of nontrivial calculations, which we defer to Appendix A. □

**Remark.** Theorem 2.1 establishes the existence of one equilibrium outcome attainable under no-external-regret learning. It does not imply that an arbitrary instantiation of a no-external-regret learner will exhibit this behavior. Establishing such learner-specific guarantees requires assumptions on the learning dynamics, namely how prices are explored and updated over time, and cannot be derived from the no-external-regret property alone. Dynamics-based analyses under additional behavioral or algorithmic assumptions are an active direction, pursued for example by Arunachaleswaran et al. (2024) and Bichler et al. (2024).

We next focus on correlated equilibrium. Jann & Schottmüller (2015) previously observed in the continuous Bertrand pricing game that, for an arbitrary non-increasing demand function, each player's expected utility goes to zero. It is therefore natural to expect a similar phenomenon in the discrete setting, and we show that this is indeed the case in the following theorem.

**Theorem 2.2.** *Consider any non-increasing demand function $f : \mathcal{P} \to [0, 1]$ and any correlated equilibrium $\mathcal{D}$. For each player $i \in \{1, 2\}$, the following holds:*

- *If $c < 1$, then $\mathbb{E}_{x \sim \mathcal{D}}[u_i(x)] \in \left[0, \frac{f(c+1/k)}{k}\right]$.*

- *If $c = 1$, then $\mathbb{E}_{x \sim \mathcal{D}}[u_i(x)] = 0$.*

*In other words, for any non-increasing demand function, any pair of no-swap-regret learners drive prices down to the marginal cost when competing against each other.*

*Proof Sketch.* At a high level, assuming $f(c + 1/k) > 0$, we argue that whenever a player chooses a price $p > c+2/k$ and the other player responds with a price at most $p$, then the former player can swap $p$ for a lower price and obtain higher utility. This creates downward pressure on prices, and the theorem follows. For a formal proof, we refer the reader to Appendix A. $\square$

Recall that correlated equilibrium is the special case of $\Phi$-correlated equilibrium in which, for both players, $\Phi_i$ contains all deviation maps. Since we showed above that under any CE the players obtain only low utility, it is natural to ask whether this conclusion continues to hold if we weaken the deviation class for one player, allowing that player's $\Phi_i$ to contain only constant maps while the other player's $\Phi_i$ still contains all deviation maps. Unfortunately, this is not the case, as we show in the following theorem.

**Theorem 2.3.** *Consider marginal cost $0 \leq c < 1$ such that $1 - c$ is a constant independent of $k$. For the constant demand function $f(x) = 1$, there exists a $\Phi$-correlated equilibrium $\mathcal{D}$ in which $\Phi_1$ contains all deviation maps and $\Phi_2$ contains only constant deviation maps such that, for each player $i \in \{1, 2\}$, the following holds:*

$$\mathbb{E}_{x \sim \mathcal{D}}[u_i(x)] \geq \lambda_0 \cdot \max_{x \in \mathcal{P}} (x - c) \, f(x),$$

*where $\lambda_0 > 0$ is an absolute constant independent of $k$.*

*In other words, there exists a demand function and a pair of learners, one no-external-regret and one no-swap-regret, such that they can sustain high transaction prices while competing against each other.*

*Proof Sketch.* Let $k_0 = k - ck$. Recall that $1 - c$ is a constant independent of $k$. Now we describe a $\Phi$-correlated

equilibrium $\mathcal{D}$ in which $\Phi_1$ contains all deviation maps and $\Phi_2$ contains only constant deviation maps. We begin by describing the marginal distribution $\mathcal{D}_1$ of the prices chosen by the player 1. Fix an integer $M := \left\lfloor \lambda_1(k_0 - 1) \right\rfloor$, where $\lambda_1$ is some small positive constant independent of $k$.

We now define $p_j$, the probability that the player 1 chooses the price $c + j/k$, as follows:

$$p_j = \begin{cases} 0 & \text{if } j < M \text{ or } j = k_0, \\ \frac{1}{M+1} & \text{if } j = M, \\ \frac{M}{j(j+1)} & \text{if } M < j < k_0 - 1, \\ \frac{M}{k_0 - 1} & \text{if } j = k_0 - 1. \end{cases}$$

The above values are nonnegative and sum to 1.

Using nontrivial calculations involving properties of harmonic numbers, we can show that $\mathcal{D}$ is indeed a $\Phi$-correlated equilibrium. The same calculations also establish the utility guarantee in the theorem statement. For details, we refer the reader to Appendix A.

$\square$

The previous result raises an important question: in an oligopoly with $n \geq 3$ players, is it sufficient that a $\Phi$-correlated equilibrium has just two players whose $\Phi_i$ contain all deviation maps to ensure that all players' expected utilities remain low, mirroring our duopoly result for CE? We answer this question in the affirmative in the following theorem.

**Theorem 2.4.** *Consider any non-increasing demand function $f : \mathcal{P} \to [0, 1]$ and any $n \geq 3$. Let $\mathcal{D}$ be a $\Phi$-correlated equilibrium such that at least two players have $\Phi_i$ containing all deviation maps. Then, for each player $i \in [n]$, the following holds:*

- *If $c < 1$, then $\mathbb{E}_{x \sim \mathcal{D}}[u_i(x)] \leq \frac{f(c+1/k)}{k}$.*

- *If $c = 1$, then $\mathbb{E}_{x \sim \mathcal{D}}[u_i(x)] \leq 0$.*

*In other words, for any non-increasing demand function, a pair of no-swap-regret learners suffices to drive the transaction price down to marginal cost, or even below it.*

The proof of the above theorem follows a similar line of reasoning to that of Theorem 2.2, and the details are provided in Appendix A.

Finally, we return to CCE and ask a central question: as the number of players $n$ grows, does the maximum achievable sum of players' expected utilities over all CCE remain a constant fraction of the maximum monopoly utility? If not,

how does this quantity decay with $n$? We answer these questions in the following theorem.

**Theorem 2.5.** *Consider a bertrand game with $n \geq 2$ players and $k \geq 5$. Then in any CCE, the total sum of expected utilities across all the players is at most*

$$\frac{4f(c+1/k)}{k} + n(1-c)f(c+1/k)e^{1-n/2}.$$

*In other words, for any non-increasing demand function, the highest utility that no-external-regret learners can attain when competing with one another decays exponentially as the number of firms increases.*

*Proof Sketch.* Let us extend the demand function $f$ to include zero by setting $f(0) = f(1/k)$. Consider a joint distribution $\mathcal{D}$ which is a CCE of the bertrand game with $n$ players. Let $X = (X_1, X_2, \ldots, X_n)$ denote the random price tuple drawn from the $\mathcal{D}$ where $X_i$ denote the price set by the player $i$. Let $u_i(X)$ denote the payoff of player $i$. Note that $\sum_{j=1}^{n} u_j(X) = (\min_{j \in [n]} X_j - c) \cdot f(\min_{j \in [n]} X_j)$. Let $U_i := \mathbb{E}[u_i(X)]$ for all $i \in [n]$ and $W := \mathbb{E}[(\min_j X_j - c) \cdot f(\min_{j \in [n]} X_j)] = \sum_{j=1}^{n} U_j$.

Consider a player $i$ and a price $c + a \in \mathcal{P}$. Let $X_{-i} := \min_{j \neq i} X_j$. Observe that $\mathbb{E}[u_i(c+a, X_{-i})] \geq a \cdot \Pr(X_{-i} > c + a) \cdot f(c + a)$. As the distribution $\mathcal{D}$ is a CCE, we have $U_i \geq \mathbb{E}[u_i(c + a, X_{-i})]$. Therefore, we have $\Pr(X_{-i} > c + a) \cdot f(c + a) \leq \min\{f(c+a), \frac{U_i}{a}\}$. Now we have the following:

$$\Pr\left(\min_{j \in [n]} X_j > c + a\right) \cdot f(c + a)$$

$$\leq \frac{1}{n} \sum_{j=1}^{n} \Pr(X_{-j} > c + a) \cdot f(c + a)$$

$$\leq \frac{1}{n} \sum_{j=1}^{n} \min\left\{f(c+a), \frac{U_j}{a}\right\}$$

$$\leq \min\left\{f(c+a), \frac{W}{na}\right\}$$

Let $\lambda := W/n$ and $k_0 = k - ck$. Using the above inequality and some nontrivial calculations, we can show the following:

$$W = \mathbb{E}[(\min_{j \in [n]} X_j - c) \cdot f(\min_{j \in [n]} X_j)]$$

$$\leq \frac{1}{k}\left(f(c+1/k) + \sum_{i=1}^{k_0-1} \min\left\{f(c+i/k), \frac{\lambda k}{i}\right\}\right)$$

$$\leq \frac{2f(c+1/k)}{k} + \lambda + \lambda \ln\left(\frac{(1-c)f(c+1/k)}{\lambda}\right)$$

Using the above inequality, and analyzing the case $W > 4f(c+1/k)/k$ with some straightforward calculations, we can prove the theorem statement. For more details, we refer the reader to Appendix A. $\square$

### 2.2. Asymmetric costs setting

We now consider the Bertrand duopoly and extend our results from the symmetric-cost setting to the asymmetric-cost setting, where player 1 has a lower marginal cost than player 2, and the gaps $c_2 - c_1$ and $1 - c_2$ are fixed positive constants, independent of $k$ and much larger than $1/k$. Recall that in the symmetric-cost setting, for any non-increasing demand function there exists a CCE under which both players obtain a constant fraction of the maximum monopoly utility. In contrast, analogous guarantees in the asymmetric-cost setting have been largely absent, even for special cases such as linear demand. This raises an important question: does there exist a CCE in the asymmetric-cost setting under which both players obtain a constant fraction of their respective maximum monopoly utilities? We answer this question in the following theorem, the formal statement of which is provided in Appendix A.

**Theorem 2.6** (Informal). *For a broad class of demand functions, including constant, linear, quadratic, and exponential demand, there exists a CCE in which both players obtain a constant fraction of their respective monopoly-optimal utilities. In other words, there exist two no-external-regret learners that can sustain high transaction prices while competing against each other under such demand functions.*

We next focus on correlated equilibrium. For asymmetric-cost setting, Jann & Schottmüller (2015) previously observed in the continuous Bertrand pricing game that, for an arbitrary non-increasing demand function, the second player which has the higher marginal cost, its expected utility is zero. We show that this conclusion continues to hold in the discrete setting in the following theorem.

**Theorem 2.7.** *Consider any non-increasing demand function $f : \mathcal{P} \to [0, 1]$, marginal costs $c_1 < c_2$ such that $c_2 - c_1 > 1/k$ and any correlated equilibrium $\mathcal{D}$. Then we have the following:*

$$\mathbb{E}_{x \sim \mathcal{D}}[u_2(x)] = 0$$

*In other words, for any non-increasing demand function, any pair of no-swap-regret learners drive player 2's utility to zero when competing against each other.*

Recall that in the symmetric-cost setting we showed that weakening the deviation class for one player, by allowing that player's $\Phi_i$ to contain only constant deviation maps while the other player's $\Phi_i$ contains all deviation maps, does not guarantee that both players obtain low utility under every $\Phi$-correlated equilibrium. This raises a natural question in

the asymmetric-cost setting: if we again restrict one player's $\Phi_i$ to contain only constant deviation maps while allowing the other player's $\Phi_i$ to contain all deviation maps, can we guarantee that player 2 receives zero expected utility under any such $\Phi$-correlated equilibrium? Surprisingly, the answer is no, as we show in the following theorem.

**Theorem 2.8.** *For the constant demand function $f(x) = 1$, there exists marginal costs $c_1 < c_2$ and a $\Phi$-correlated equilibrium $\mathcal{D}$ in which $\Phi_1$ contains all deviation maps and $\Phi_2$ contains only constant deviation maps such that, for each player $i \in \{1, 2\}$ and large $k$, the following holds:*

$$\mathbb{E}_{x \sim \mathcal{D}}[u_i(x)] \geq \lambda_0 \cdot \max_{x \in \mathcal{P}}(x - c_i)\, f(x),$$

*where $\lambda_0 > 0$ is an absolute constant independent of $k$. Similarly, for the same marginal costs $c_1 < c_2$, a $\Phi$-correlated equilibrium $\mathcal{D}$ in which $\Phi_2$ contains all deviation maps and $\Phi_1$ contains only constant deviation maps such that, for each player $i \in \{1, 2\}$ and large $k$, the following holds:*

$$\mathbb{E}_{x \sim \mathcal{D}}[u_i(x)] \geq \lambda_1 \cdot \max_{x \in \mathcal{P}}(x - c_i)\, f(x),$$

*where $\lambda_1 > 0$ is an absolute constant independent of $k$.*

# 3. Experiments

In this section, we present our experiments. In Sections 3.1 and 3.2, we numerically investigate, under both symmetric and asymmetric cost settings, how the largest expected utility attainable by a player under any CCE compares to the corresponding maximum monopoly utility across standard demand functions. We then study no-swap-regret learners in Section 3.2 and analyze the prices selected by these learners. The code for the experiments is available at https://github.com/maitiarnab9/bertrand-paradox.

## 3.1. Symmetric cost setting

In this section, we restrict attention to the symmetric cost setting and to symmetric CCE. A symmetric CCE is a CCE supported only on price tuples in which all components are equal. This restriction is sufficient for our purposes because, when costs are symmetric, any CCE can be transformed into a symmetric CCE without changing the sum of players' expected utilities. A formal proof of this claim is provided in Appendix C.2.

We begin by numerically investigating how the largest expected utility attainable by a player under any symmetric CCE compares to maximum monopoly utility across standard demand functions, namely constant ($f(x) = 1$), linear ($f(x) = 1 - x$), quadratic ($f(x) = 1 - x^2$), and exponential ($f(x) = \exp(-x)$) demand. For each $k \in \{10, 11, \ldots, 100\}$, we compute the ratio of the maxi-

mum expected utility over all symmetric CCEs to the maximum monopoly utility, and plot this ratio as a function of $k$. Figure 1a presents the results for marginal cost $c = 0$ and constant demand function. The ratio appears to converge to $1/e$ as $k$ increases. We observe the same qualitative behavior for other demand functions and cost levels, with additional plots provided in Appendix B.

We next numerically study how the largest expected utility attainable by a player under any symmetric CCE scales with the number of firms $n$ across standard demand functions, namely constant, linear, quadratic, and exponential demand. For each $n \in \{2, 3, \ldots, 10\}$, we compute the ratio of the maximum expected utility over all symmetric CCEs to the maximum monopoly utility, and plot this ratio as a function of $n$ with $k$ fixed to 100. Figure 1b presents the results for marginal cost $c = 0$ and constant demand function. The ratio decays exponentially in $n$. We observe the same qualitative behavior for other demand functions and cost levels, with additional plots provided in Appendix B.

## 3.2. Asymmetric cost setting

In this section, we study the asymmetric-cost setting in a Bertrand duopoly with $c_1 = 0$ and $c_2 > 0$. We begin by numerically investigating, across standard demand functions (constant, linear, quadratic, and exponential), how the largest expected utility attainable by each player under a CCE compares to the corresponding maximum monopoly utility.

We first consider a CCE that maximizes the expected utility of player 1. For this equilibrium, we compute, for each player, the ratio between the player's expected utility and the corresponding maximum monopoly utility, and plot these ratios as functions of $c_2$. Figure 1c presents the results for constant demand with $k = 100$. As $c_2$ increases, player 1's ratio increases while player 2's ratio decreases. We observe the same qualitative behavior for the other demand functions, with additional plots provided in Appendix B.

We next consider a CCE that maximizes the expected utility of player 2. We again compute the two players' utility ratios and plot them as functions of $c_2$. Figure 1d presents the results for constant demand with $k = 100$. Somewhat surprisingly, both ratios remain approximately constant as $c_2$ increases, and only begin to diverge for larger values of $c_2$ approaches 1. We observe a similar pattern for the other demand functions, with additional plots in Appendix B.

We next turn to correlated equilibria and study which prices emerge under CEs induced by different no-swap-regret learners. To illustrate this behavior, we focus on the constant-demand function. We consider two families of no-swap-regret learners: (i) regret matching (Hart & Mas-Colell, 2000), and (ii) an external-to-internal reduction-

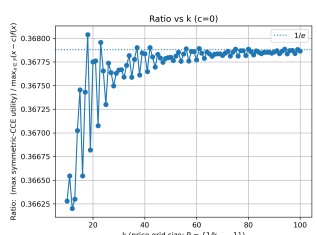 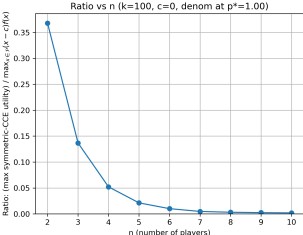 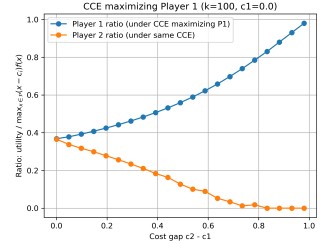 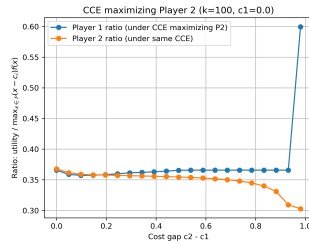

*(a)* Ratio of duopoly utility under the best symmetric CCE and monopoly utility    *(b)* Exponential decay of utility under the best symmetric CCE    *(c)* Utility Ratios under the CCE that maximizes player 1's utility    *(d)* Utility Ratios under the CCE that maximizes player 2's utility

*Figure 1.* Numerical experiments for constant demand with $c_1 = 0$.

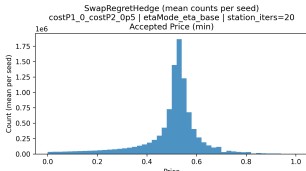 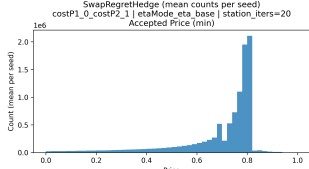 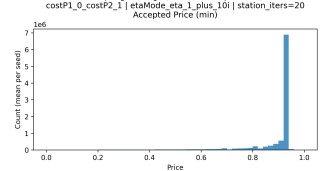 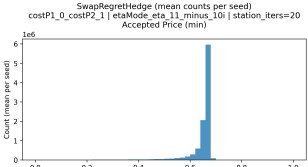

*(a)* $c_2 = 0.5$ with same learning rates for both players    *(b)* $c_2 = 1$ with same learning rates for both players    *(c)* $c_2 = 1$ with larger learning rate for the second player    *(d)* $c_2 = 1$ with larger learning rate for the first player

*Figure 2.* Experiments using the Hedge-based no-swap-regret learner with different combinations of costs and learning rates. The plots show the frequency of transaction prices over $T = 10^7$ rounds, averaged across 100 random seeds.

based learner that invokes Hedge as a subroutine (Blum & Mansour, 2007). Each learner's action set is the discretized price grid $\mathcal{P} = \{1/k, 2/k, \ldots, 1\}$ with $k = 100$. In our experiments, we fix a learner class and have both players run an algorithm from that class for $T = 10^7$ rounds to select prices. At round $t \in [T]$, player 1 selects price $i_t/k$ and player 2 selects price $j_t/k$. After observing the opponent's price, each player receives full-information feedback given by its entire utility vector against the realized opponent action. Concretely, player 1 observes $u_1(\cdot, j_t/k)$ and player 2 observes $u_2(i_t/k, \cdot)$, and supplies this feedback to its respective learner.

While our theoretical result implies that player 2 receives zero utility in any CE, there exist CEs in which player 1 attains utility $c_2$. This raises the question of whether standard no-swap-regret learners reliably converge to such favorable equilibria for player 1. To investigate this, we consider two cost values, $c_2 = 0.5$ and $c_2 = 1$. Aggregating over 100 random seeds, Figure 2 shows that when both players use the Hedge-based no-swap-regret learner with identical learning rates, the most frequently chosen price for player 1 is close to 0.5 when $c_2 = 0.5$. In contrast, when $c_2 = 1$, the most frequent price is not close to 1, but instead around 0.8. Moreover, this most frequently chosen price increases when we increase player 2's learning rate and decreases when we increase player 1's learning rate. We observe a similar pattern for regret matching, with the corresponding experimental results deferred to Appendix B. Appendix B

also provides a more detailed explanation for the surprising behavior when $c_2 = 1$, which does not appear at $c_2 = 0.5$.

Overall, these experiments suggest that there is no universal answer to the question above. The particular CE to which no-swap-regret learning dynamics converge can depend sensitively on the cost parameters and on algorithmic details such as learning rates. This motivates analysis beyond generic convergence guarantees for no-swap-regret learners and points to an important open direction.

## 4. Conclusion

In this paper, we study equilibrium outcomes attainable by no-regret learners in Bertrand pricing game. In the symmetric-cost setting, we show the existence of two no-external-regret learners that can sustain high transaction prices. In contrast, when both firms incur low swap regret, transaction prices are driven down; moreover, a single no-swap-regret learner need not be sufficient to induce low prices. We also extend our results to the asymmetric duopoly and provide experiments that support the theory and reveal additional, unexpected phenomena.

Our work raises several open questions. Which properties of no-external-regret learners determine whether high transaction prices are sustained, beyond the regret guarantee itself? Given a concrete instantiation of a no-swap-regret learner, can one always construct a no-external-regret learner that sustains high transaction prices against it? Finally, in the

asymmetric duopoly with $c_1 < c_2$, what properties of no-swap-regret learners drive transaction prices far below $c_2$?

## Impact Statement

This paper presents work whose goal is to advance the field of Machine Learning. There are many potential societal consequences of our work, none which we feel must be specifically highlighted here.

## ACKNOWLEDGEMENTS

LJ Ratliff was supported in part by NSF 1844729, 2312775. KJ and AM were supported in part by a Microsoft Grant for Customer Experience Innovation and a Singapore AI Visiting Professorship award.

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

## A. Omitted proofs

*Proof of Theorem 2.1.* We now describe a way to construct a symmetric CCE where both players always choose the same price. Let $s_i := (i/k - c) \cdot f(i/k)$ and $S_i := \max_{j \leq i} s_j$. Let $s_{\max} := \max_{i \in [k]} s_i$ and consider the smallest index $m$ such that $m \in \arg\max_{i \in [k]} s_i$. Consider a constant $\lambda := \frac{1}{2e^2}$ and set $B := \lambda s_{\max}$. We now consider the following two cases.

**Case 1: $s_{k \cdot c + 1} \geq B$.** We choose the price $c + 1/k$ for both the players with probability 1. Note that this is actually a Nash equilibrium as no player can unilaterally deviate and get a higher reward. Now observe that each player receives a utility of $\frac{1}{2} s_{k \cdot c + 1} \geq B/2 = (\lambda/2) \cdot s_{\max}$.

**Case 2: $s_{k \cdot c + 1} < B$.** Let $i_0 := \min\{i : S_i \geq B\}$. We randomly choose a price $x = i/k$ for both the players with probability $\tau_i - \tau_{i+1}$, where $\tau_i$ is defined as follows:

$$\tau_i := \Pr[x \geq i/k] = \begin{cases} 1, & i < i_0, \\ B/S_i, & i_0 \leq i \leq m, \\ 0, & i > m, \end{cases}$$

Now we prove that we have indeed constructed a symmetric CCE. Due to symmetry, showing that $\mathbb{E}[u_1(i/k, x)] \leq \mathbb{E}[u_1(x, x)]$ for any $i \in [k]$ suffices, where $x$ is the price randomly chosen for both the players as per the distribution above. First, we have the following:

$$\mathbb{E}[u_1(i/k, x)] = s_i \cdot \left(\Pr[x > i/k] + \tfrac{1}{2}\Pr[x = i/k]\right) \leq s_i \tau_i \leq B.$$

We get the last inequality due to the fact that if $i < i_0$, then $s_i < B$ and if $i \geq i_0$ then $s_i \tau_i \leq S_i \cdot \frac{B}{S_i} = B$.

Consider $i > kc$. As $f$ is non-increasing, we have $k \cdot s_i/(i - kc) = f(i/k) \geq f((i+1)/k) = k \cdot s_{i+1}/(i+1-kc)$. Hence if $S_i < S_{i+1}$, then $S_{i+1} = s_{i+1} \leq \frac{i+1-kc}{i-kc} \cdot s_i \leq 2s_i$. This also implies that $S_{i_0} \leq 2B$. Now, let $\mathcal{I} := \{i \in \{i_0, i_0 + 1, \ldots, m-1\} : S_i < S_{i+1}\}$. For any $i \in \mathcal{I}$, we have the following

$$s_i \cdot \left(\frac{1}{S_i} - \frac{1}{S_{i+1}}\right) = \frac{s_i}{S_{i+1}} \cdot \frac{S_{i+1} - S_i}{S_i} \geq \tfrac{1}{2} \cdot \frac{S_{i+1} - S_i}{S_i}.$$

Now, we have the following:

$$\begin{aligned}
\sum_{i \in \mathcal{I}} \frac{S_{i+1} - S_i}{S_i} &= \sum_{i \in \mathcal{I}} \left(\frac{S_{i+1}}{S_i} - 1\right) \\
&\geq \sum_{i \in \mathcal{I}} \ln \frac{S_{i+1}}{S_i} && \text{(as } \ln r \leq r - 1 \text{ for all } r > 0\text{)} \\
&= \ln \frac{S_m}{S_{i_0}} && \text{(as } \min_{i \in \mathcal{I}} S_i = S_{i_0} \text{ and } m - 1 \in \mathcal{I}\text{)} \\
&= \ln \frac{s_{\max}}{S_{i_0}} \\
&\geq \ln \frac{1}{2\lambda} = 2. && \text{(as } B = \lambda s_{\max} \text{ and } S_{i_0} \leq 2B\text{)}
\end{aligned}$$

Hence, we have the following:

$$\begin{aligned}
\mathbb{E}[u_1(x, x)] &\geq \frac{1}{2} \sum_{i=i_0}^{m-1} s_i \cdot \left(\frac{B}{S_i} - \frac{B}{S_{i+1}}\right) + \frac{1}{2} s_m \cdot \frac{B}{S_m} \\
&\geq \frac{B}{2} \sum_{i \in \mathcal{I}} s_i \cdot \left(\frac{1}{S_i} - \frac{1}{S_{i+1}}\right) + B/2 && \text{(as } S_m = s_m\text{)} \\
&\geq B/2 + B/2 && \text{(due to the above calculations)} \\
&= B
\end{aligned}$$

This concludes the proof of this theorem. $\qquad\square$

We next prove the following technical lemma.

**Lemma A.1.** *Consider a player $i$ and a non-increasing demand function $f$ such that $f\left(c + \frac{1}{k}\right) > 0$. Let $\mathcal{D}$ be a distribution over price tuples $x_{-i} \in \mathcal{P}^{n-1}$, and for any tuple $x_{-i}$ let $\min(x_{-i})$ denote its minimum component. Let $\frac{i_\star}{k}$ be the largest value of $\min(x_{-i})$ among all tuples $x_{-i}$ in the support of $\mathcal{D}$, and assume $i_\star > ck$. Define*

$$\mathcal{P}_\star := \left\{ \frac{i_\star}{k}, \frac{i_\star + 1}{k}, \dots, 1 \right\} \setminus \left\{ c + \frac{1}{k}, c + \frac{2}{k} \right\}.$$

*Then*

$$\max_{x_i \in \mathcal{P} \setminus \mathcal{P}_\star} \mathbb{E}_{x_{-i} \sim \mathcal{D}}[u_i(x_i, x_{-i})] > \max_{x_i \in \mathcal{P}_\star} \mathbb{E}_{x_{-i} \sim \mathcal{D}}[u_i(x_i, x_{-i})].$$

*Proof.* Observe that $\mathbb{E}_{x_{-i} \sim \mathcal{D}}[u_i(c + 1/k, x_{-i})] > 0$ and $\mathbb{E}_{x_{-i} \sim \mathcal{D}}[u_i(i/k, x_{-i})] = 0$ for all $i > i_\star$. Next we have the following if $i_\star/k \geq c + 3/k$ and $f\left(\frac{i_\star - 1}{k}\right) > 0$:

$$\begin{aligned}
\mathbb{E}_{x_{-i} \sim \mathcal{D}}\left[u_i\left(\frac{i_\star - 1}{k}, x_{-i}\right)\right] &\geq \left(\frac{i_\star - 1}{k} - c\right) \cdot f\left(\frac{i_\star - 1}{k}\right) \cdot \mathbb{P}_{x_{-i} \sim \mathcal{D}}[\min(x_{-i}) = i_\star/k] \\
&> \frac{1}{2} \cdot \left(\frac{i_\star}{k} - c\right) \cdot f\left(\frac{i_\star}{k}\right) \cdot \mathbb{P}_{x_{-i} \sim \mathcal{D}}[\min(x_{-i}) = i_\star/k] \\
&\geq \mathbb{E}_{x_{-i} \sim \mathcal{D}}\left[u_i\left(\frac{i_\star}{k}, x_{-i}\right)\right]
\end{aligned}$$

If $f\left(\frac{i_\star - 1}{k}\right) = 0$, then $\mathbb{E}_{x_{-i} \sim \mathcal{D}}[u_i(i_\star/k, x_{-i})] = 0$ whereas $\mathbb{E}_{x_{-i} \sim \mathcal{D}}[u_i(c + 1/k, x_{-i})] > 0$.

$\square$

We next prove Theorem 2.2.

*Proof of Theorem 2.2.* Let $\mathcal{D}$ be a correlated equilibrium. First observe that for any pair of prices $(x_1, x_2)$ which has a positive weight under the CE $\mathcal{D}$, we have $\min\{x_1, x_2\} \geq c$ otherwise the player setting the lowest price can swap their price with $c$ and get a higher utility. If $c = 1$ then a weight of one on $(c, c)$ is the unique CE and $\mathbb{E}_{x \sim \mathcal{D}}[u_i(x)] = 0$ for any $i \in \{1, 2\}$.

Now let us consider the case where $c < 1$. If $f(c + 1/k) = 0$, then again we get $\mathbb{E}_{x \sim \mathcal{D}}[u_i(x)] = 0$ for any $i \in \{1, 2\}$. Hence let us assume that $f(c + 1/k) > 0$.

Now observe that for any pair of prices $(x_1, x_2)$ which has a positive weight under the CE $\mathcal{D}$ and $\min\{x_1, x_2\} = c$, then $x_1 = x_2 = c$ otherwise the player setting the price $c$ can swap their price with $c + 1/k$ and get a higher utility. If $(c, c)$ has a weight of one under $\mathcal{D}$, then it forms a CE and $\mathbb{E}_{x \sim \mathcal{D}}[u_i(x)] = 0$ for any $i \in \{1, 2\}$.

Let us now assume that $(c, c)$ does not have a weight of one under $\mathcal{D}$. Now we claim that for any pair of prices $(x_1, x_2)$ which has a positive weight under the CE $\mathcal{D}$, we have $\max\{x_1, x_2\} \leq c + 2/k$. For the sake of contradiction, let us assume that this is not true. Let $i_1/k$ be the largest price that the player 1 chooses under $\mathcal{D}$ and let $i_2/k$ be the largest price that the player 2 chooses under $\mathcal{D}$. W.l.o.g let us assume that $i_1 \geq i_2$. In this case we have $i_1/k \geq c + 3/k$. Now due to Lemma A.1, we have the following:

$$\mathbb{E}_{(x_1, x_2) \sim \mathcal{D}}[u_1(i_1/k, x_2)|x_1 = i_1/k] < \max_{x_1' \in \mathcal{P}} \mathbb{E}_{(x_1, x_2) \sim \mathcal{D}}[u_1(x_1', x_2)|x_1 = i_1/k]$$

This is a contradiction to our assumption that $\mathcal{D}$ is a correlated equilibrium. Hence, for any pair of prices $(x_1, x_2)$ which has a positive weight under the CE $\mathcal{D}$, we have $\max\{x_1, x_2\} \leq c + \frac{2}{k}$.

Now we claim that for any pair of prices $(x_1, x_2)$ which has a positive weight under the CE $\mathcal{D}$, we have $x_1 = x_2$. We already proved the claim for the case when $\min\{x_1, x_2\} = c$. Also since $\max\{x_1, x_2\} \leq c + \frac{2}{k}$, if $\min\{x_1, x_2\} = c + \frac{2}{k}$, then $x_1 = x_2 = c + \frac{2}{k}$. Now let us consider the case when $\min\{x_1, x_2\} = c + \frac{1}{k}$. If $\max\{x_1, x_2\} = c + \frac{2}{k}$, then player

setting price of $c + \frac{2}{k}$ can lower its price to $c + 1/k$ and get a higher reward. Hence, in this case as well $x_1 = x_2$. If $\lambda_1$ is the weight on $(c + 1/k, c + 1/k)$ and $\lambda_2$ is the weight on $(c + 2/k, c + 2/k)$, then for any $i \in \{1, 2\}$ we have the following:

$$\mathbb{E}_{x \sim \mathcal{D}}[u_i(x)] = \frac{\lambda_1}{2k} \cdot f(c + 1/k) + \frac{\lambda_2}{k} \cdot f(c + 2/k) \in \left[0, \frac{f(c + 1/k)}{k}\right].$$

$\square$

*Proof of Theorem 2.3.* Let $k_0 = k - ck$. Recall that $1 - c$ is a constant independent of $k$. Let us now assume that $k_0 \geq 3e^{10} + 1$ as the other case is trivial (just choose $c + 1/k$ for both players). Now we describe a $\Phi$-correlated equilibrium $\mathcal{D}$ in which $\Phi_1$ contains all deviation maps and $\Phi_2$ contains only constant deviation. We begin by describing the marginal distribution $\mathcal{D}_1$ of the prices chosen by the player 1. Fix an integer $M := \left\lfloor \frac{k_0 - 1}{e^{10}} \right\rfloor$.

We now define $p_j$, the probability that the player 1 chooses the price $c + j/k$, as follows:

$$p_j = \begin{cases} 0 & \text{if } j < M \text{ or } j = k_0, \\ \frac{1}{M+1} & \text{if } j = M, \\ \frac{M}{j(j+1)} & \text{if } M < j < k_0 - 1, \\ \frac{M}{k_0 - 1} & \text{if } j = k_0 - 1. \end{cases}$$

The above values are nonnegative and sum to 1 which we prove later.

Next, we describe how player 2 selects a price conditional on the price chosen by player 1. Given that player 1 chooses the price $c + i/k$, player 2 chooses the price $c + (i+1)/k$ with probability $1/2$ and the price $c + \lfloor i/2 \rfloor /k$ with probability $1/2$.

First, we show that the player 1 has low swap regret. Consider a price $c + i/k$ such that $p_i > 0$. First, observe that $\mathbb{E}_{(x_1, x_2) \sim \mathcal{D}}[u_1(x_1, x_2)|x_1 = c + i/k] = i/(2k)$. Next observe that $\mathbb{E}_{(x_1, x_2) \sim \mathcal{D}}[u_1(c + (i+1)/k, x_2)|x_1 = c + i/k] = (i+1)/(4k) \leq i/(2k)$. Next observe that $\mathbb{E}_{(x_1, x_2) \sim \mathcal{D}}[u_1(c + j/k, x_2)|x_1 = c + i/k] = 0$ for all $j > i + 1$. Next observe that $\mathbb{E}_{(x_1, x_2) \sim \mathcal{D}}[u_1(c + j/k, x_2)|x_1 = c + i/k] = j/(2k) < i/(2k)$ for all $\lfloor i/2 \rfloor < j < i$. Finally, observe that $\mathbb{E}_{(x_1, x_2) \sim \mathcal{D}}[u_1(c + j/k, x_2)|x_1 = c + i/k] \leq i/(2k)$ for all $j \leq \lfloor i/2 \rfloor$. Hence, the player 1 has low swap regret.

Next, we show that the player 2 has low external regret.

Let $S_i := \sum_{j \geq i} p_j$. We have the following for all $i \in [k_0 - 1]$ (which we prove later):

$$S_i = \min\left(1, \frac{M}{i}\right),$$

so

$$\mathbb{E}_{x \sim \mathcal{D}_1}[x - c] = \frac{1}{k} \sum_{j=1}^{k_0 - 1} j p_j = \frac{1}{k} \sum_{j=1}^{k_0 - 1} \sum_{i=1}^{j} p_j = \frac{1}{k} \sum_{i=1}^{k_0 - 1} \sum_{j=i}^{k_0 - 1} p_j = \frac{1}{k} \sum_{i=1}^{k_0 - 1} S_i$$

$$= \frac{1}{k}\left(M + M \sum_{i=M+1}^{k_0 - 1} \frac{1}{i}\right) = \frac{M}{k}\left(1 + H_{k_0 - 1} - H_M\right),$$

where $H_n$ is the $n$-th harmonic number.

Now we have the following for any $i \in [k_0]$:

$$\mathbb{E}_{x_1 \sim \mathcal{D}_1}[u_2(x_1, c + i/k)] = \frac{i}{k}\left(\mathbb{P}_{x_1 \sim \mathcal{D}_1}(x_1 > c + i/k) + \frac{1}{2}\mathbb{P}_{x_1 \sim \mathcal{D}_1}(x_1 = c + i/k)\right) \leq \frac{i}{k} S_i.$$

Hence,

$$\max_{i \in [k_0]} \mathbb{E}_{x_1 \sim \mathcal{D}_1}[u_2(x_1, c + i/k)] \leq \frac{1}{k} \max_{i \in [k_0]} i \cdot S_i = \frac{M}{k}.$$

Due to the properties of Harmonic number, we have $H_{k_0-1} - H_M \geq \ln\frac{k_0-1}{M} - \frac{1}{2M}$ (which we prove later). With $M = \lfloor (k_0 - 1)/e^{10} \rfloor$, we have $H_{k_0-1} - H_M \geq 9.5$, so

$$\frac{\mathbb{E}_{x \sim \mathcal{D}_1}[x - c]}{\max_{i \in [k_0]} \mathbb{E}_{x_1 \sim \mathcal{D}_1}[u_2(x_1, c + i/k)]} \geq 1 + H_{k_0-1} - H_M \geq 9.5.$$

Next we have the following:

$$\mathbb{E}_{(x_1,x_2) \sim \mathcal{D}}[u_2(x_1, x_2)] = \frac{1}{2} \cdot \sum_{i=M}^{k_0-1} p_i \cdot \left( \frac{\lfloor i/2 \rfloor}{k} \right)$$

$$\geq \frac{1}{2} \cdot \sum_{i=M}^{k_0-1} p_i \cdot \left( \frac{i}{4k} \right)$$

$$= (1/8) \cdot \mathbb{E}_{x \sim \mathcal{D}_1}[x - c]$$

$$\geq \max_{i \in [k_0]} \mathbb{E}_{x_1 \sim \mathcal{D}_1}[u_2(x_1, c + i/k)]$$

Hence, player 2 has a low external regret.

Also observe that $\mathbb{E}_{(x_1,x_2) \sim \mathcal{D}}[u_1(x_1, x_2)] \geq (1/2) \cdot \mathbb{E}_{x \sim \mathcal{D}_1}[x - c]$. Hence, in order to prove the theorem, it suffices to show that $\mathbb{E}_{x \sim \mathcal{D}_1}[x - c]$ is large. Towards that we have the following:

$$\mathbb{E}_{x \sim \mathcal{D}_1}[x - c] = \frac{M}{k}\left(1 + H_{k_0-1} - H_M\right) \geq \frac{4.74(1 - c)}{e^{10}}.$$

**Omitted calculations**

First, we prove that $\sum_{j=1}^{k_0} p_j = 1$. Using the identity $\frac{1}{j(j+1)} = \frac{1}{j} - \frac{1}{j+1}$, we have the following:

$$\sum_{j=1}^{k_0} p_j = \frac{1}{M+1} + \sum_{j=M+1}^{k_0-2} \frac{M}{j(j+1)} + \frac{M}{k_0 - 1}$$

$$= \frac{1}{M+1} + \sum_{j=M+1}^{k_0-2} \left( \frac{M}{j} - \frac{M}{j+1} \right) + \frac{M}{k_0 - 1}$$

$$= \frac{1}{M+1} + \frac{M}{M+1} - \frac{M}{k_0 - 1} + \frac{M}{k_0 - 1}$$

$$= 1$$

Next, we prove that $S_i := \sum_{j \geq i} p_j = \left(1, \frac{M}{i}\right)$ for all $i \in [k_0 - 1]$. We divide our analysis into three cases.

Case 1: $i \leq M$

As $p_j = 0$ for all $j < M$, we have $\sum_{j<i} p_j = 0$. Therefore

$$S_i = \sum_{j \geq i} p_j = 1 = \min(1, M/i).$$

Case 2: $M + 1 \leq i \leq k_0 - 2$

In this case, we have

$$S_i = \sum_{j=i}^{k_0-2} \frac{M}{j(j+1)} + \frac{M}{k_0 - 1} = \sum_{j=i}^{k_0-2} \left( \frac{M}{j} - \frac{M}{j+1} \right) + \frac{M}{k_0 - 1} = M\left( \frac{1}{i} - \frac{1}{k_0 - 1} \right) + \frac{M}{k_0 - 1} = \frac{M}{i} = \min(1, M/i).$$

Case 3: $i = k_0 - 1$

In this case, we have

$$S_{k_0-1} = p_{k_0-1} = \frac{M}{k_0 - 1} = \frac{M}{i} = \min(1, M/i).$$

By combining all the three cases, we get $S_i = \min(1, M/i)$ for all $i \in [k_0 - 1]$.

Finally, we show that $H_{k_0-1} - H_M \geq \ln\frac{k_0-1}{M} - \frac{1}{2M}$. For any $n \geq 1$, we have the following:

$$\gamma + \ln n < H_n < \gamma + \ln n + \frac{1}{2n},$$

where $\gamma$ is the euler's constant and the inequality follows from Guo & Qi (2011).

Hence, we have the following:

$$H_{k_0-1} - H_M \geq \gamma + \ln(k_0 - 1) - (\gamma + \ln M + \frac{1}{2M}) = \ln\frac{k_0 - 1}{M} - \frac{1}{2M}$$

$\square$

*Proof of Theorem 2.4.* Let $\mathcal{D}$ be a $\Phi$-correlated equilibrium such that $\Phi_1$ and $\Phi_2$ contain all deviation maps. First, observe that if $c = 1$ then for each player $i \in [n]$ we have $\mathbb{E}_{x\sim\mathcal{D}}[u_i(x)] \leq 0$. Next, observe that if $f(c + 1/k) = 0$, then for each player $i \in [n]$ we have $\mathbb{E}_{x\sim\mathcal{D}}[u_i(x)] \leq 0$.

Let us now assume that $f(c + 1/k) > 0$. We now prove that for any tuple of prices $x$ which has a positive weight under $\mathcal{D}$, we have $\min_{i\in[n]} x_i \leq c + \frac{2}{k}$. For the sake of contradiction, let us assume that this is not true. Among the tuples that have a positive weight under $\mathcal{D}$, let $\tilde{x}$ be the tuple that maximizes $\min_{i\in[n]} x_i$. W.l.o.g let us assume that $\tilde{x}_1 \geq \tilde{x}_2$. Observe that we have $\tilde{x}_1 \geq c + 3/k$. By applying the Lemma A.1 on the marginal distribution of $x_{-1}$ conditioned on $x_1 = \tilde{x}_1$ under $\mathcal{D}$, we have the following:

$$\mathbb{E}_{x\sim\mathcal{D}}[u_1(\tilde{x}_1, x_{-1})|x_1 = \tilde{x}_1] < \max_{x_1'\in\mathcal{P}} \mathbb{E}_{x\sim\mathcal{D}}[u_1(x_1', x_{-1})|x_1 = \tilde{x}_1]$$

This is a contradiction to our assumption that $\mathcal{D}$ is a $\Phi$-correlated equilibrium. Hence, for any tuple of prices $x$ which has a positive weight under $\mathcal{D}$, we have $\min_{i\in[n]} x_i \leq c + \frac{2}{k}$.

If $f(c + 2/k) = 0$, then for each player $i \in [n]$ we have $\mathbb{E}_{x\sim\mathcal{D}}[u_i(x)] \leq \frac{f(c+1/k)}{k}$. Let us now assume that $f(c + 2/k) > 0$.

Consider tuple of prices $x$ which has a positive weight under $\mathcal{D}$. Now we claim that if $\min_{i\in[n]} x_i = c + \frac{2}{k}$, then $x_1 = x_2 = c + \frac{2}{k}$. For contradiction, let us assume that this is not true. W.l.o.g let us assume that $x_1 \geq x_2$. Observe that $x_1 > c + \frac{2}{k}$. Since for any tuple $x'$ which has a positive weight under $\mathcal{D}$ has $\min_{i\in[n]} x_i' \leq c + \frac{2}{k}$, we have the following:

$$\mathbb{E}_{x'\sim\mathcal{D}}[u_1(x_1, x_{-1}')|x_1' = x_1] = 0 < \mathbb{E}_{x\sim\mathcal{D}}[u_1(c + 2/k, x_{-1}')|x_1' = x_1].$$

This implies for each player $i \in [n]$ we have $\mathbb{E}_{x\sim\mathcal{D}}[u_i(x)] \leq \frac{2}{k} \cdot \frac{f(c+1/k)}{2} = \frac{f(c+1/k)}{k}$. $\square$

*Proof of Theorem 2.5.* Let us extend the demand function $f$ to include zero by setting $f(0) = f(1/k)$. Consider a joint distribution $\mathcal{D}$ which is a CCE of the bertrand game with $n$ players. Let $X = (X_1, X_2, \ldots, X_n)$ denote the random price tuple drawn from the $\mathcal{D}$ where $X_i$ denote the price set by the player $i$. Let $u_i(X)$ denote the payoff of player $i$. Note that $\sum_{j=1}^n u_j(X) = (\min_{j\in[n]} X_j - c) \cdot f(\min_{j\in[n]} X_j)$. Let $U_i := \mathbb{E}[u_i(X)]$ for all $i \in [n]$ and $W := \mathbb{E}[(\min_j X_j - c) \cdot f(\min_{j\in[n]} X_j)] = \sum_{j=1}^n U_j$.

Consider a player $i$ and a price $c + a \in \mathcal{P}$. Let $X_{-i} := \min_{j\neq i} X_j$. Observe that $\mathbb{E}[u_i(c + a, X_{-i})] \geq a \cdot \Pr(X_{-i} > c + a) \cdot f(c + a)$. As the distribution $\mathcal{D}$ is a CCE, we have $U_i \geq \mathbb{E}[u_i(c + a, X_{-i})]$. Therefore, we have $\Pr(X_{-i} > c + a) \cdot f(c + a) \leq \min\{f(c + a), \frac{U_i}{a}\}$. Now we have the following:

$$\Pr\left(\min_{j\in[n]} X_j > c + a\right) \cdot f(c+a) \leq \frac{1}{n} \sum_{j=1}^n \Pr(X_{-j} > c+a) \cdot f(c+a) \leq \frac{1}{n} \sum_{j=1}^n \min\left\{f(c+a), \frac{U_j}{a}\right\} \leq \min\left\{f(c+a), \frac{W}{na}\right\}$$

Let $\lambda := W/n$ and $k_0 = k - ck$. Now we have the following:

$$
\begin{aligned}
W = \mathbb{E}[(\min_{j\in[n]} X_j - c) \cdot f(\min_{j\in[n]} X_j)] &\leq \frac{1}{k} \cdot \mathbb{E}\left[\sum_{i=0}^{k_0-1} \mathbb{1}\left\{\min_{j\in[n]} X_j > c + i/k\right\} \cdot f(\min_{j\in[n]} X_j)\right] \\
&\leq \frac{1}{k} \sum_{i=0}^{k_0-1} \Pr(\min_{j\in[n]} X_j > c + i/k) \cdot f(c + (i+1)/k) \\
&\leq \frac{1}{k}\left(f(c+1/k) + \sum_{i=1}^{k_0-1} \min\left\{f(c+i/k), \frac{\lambda k}{i}\right\}\right) \\
&\leq \frac{\lfloor \lambda k/f(c+1/k)\rfloor + 2}{k} \cdot f(c+1/k) + \lambda \sum_{i=\lfloor \lambda k/f(c+1/k)\rfloor + 2}^{k_0-1} \frac{1}{i} \\
&\leq \frac{2f(c+1/k)}{k} + \lambda + \lambda \ln\left(\frac{k_0-1}{\lfloor \lambda k/f(c+1/k)\rfloor + 1}\right) \\
&\leq \frac{2f(c+1/k)}{k} + \lambda + \lambda \ln\left(\frac{k_0 f(c+1/k)}{\lambda k}\right) \\
&\leq \frac{2f(c+1/k)}{k} + \lambda + \lambda \ln\left(\frac{(1-c)f(c+1/k)}{\lambda}\right)
\end{aligned}
$$

If $W \leq 4f(c+1/k)/k$, the theorem statement holds trivially. Let us assume that $W > \frac{4f(c+1/k)}{k}$. Now, we have

$$
\frac{W}{2} \leq \lambda + \lambda \ln \frac{(1-c)f(c+1/k)}{\lambda} = \frac{W}{n}\left(1 + \ln \frac{n(1-c)f(c+1/k)}{W}\right).
$$

Now if we divide both sides by $W/2$, we have:

$$
1 \leq \frac{2}{n}\left(1 + \ln \frac{n(1-c)f(c+1/k)}{W}\right) \quad \Rightarrow \quad n/2 \leq 1 + \ln \frac{n(1-c)f(c+1/k)}{W}.
$$

Hence

$$
\ln \frac{n(1-c)f(c+1/k)}{W} \geq n/2 - 1 \quad \Rightarrow \quad \frac{n(1-c)f(c+1/k)}{W} \geq e^{n/2-1} \quad \Rightarrow \quad W \leq n(1-c)f(c+1/k)e^{1-n/2}.
$$

This concludes the proof of the theorem. $\qquad \square$

We now focus on the asymmetric-cost setting. We consider costs $c_1 < c_2$ such that the gaps $c_2 - c_1$ and $1 - c_2$ are fixed positive constants, independent of $k$. We next state Theorem 2.6 formally, including the required conditions on the demand function; these conditions are satisfied by a broad range of demand functions, including constant, linear, quadratic, and exponential demand.

**Theorem A.2** (Formal statement of Theorem 2.6). *Let $\lambda_1 \in (0,1)$ be a constant independent of $k$. Let the following conditions hold: $1/k \cdot f(c_1 + 1/k) < \frac{\lambda_1}{8e^2} \cdot \max_{x\in\mathcal{P}}(x - c_1)f(x)$, $\max_{x\in\mathcal{P}}(x - c_2)f(x) \geq \lambda_1 \cdot \max_{x\in\mathcal{P}}(x - c_1)f(x)$, and $\max_{i\in[kc_2-1]}(i/k - c_1) \cdot f(i/k) \geq \lambda_1/(4e^2) \cdot \max_{x\in\mathcal{P}}(x - c_1)f(x)$. Then there exists a CCE $\mathcal{D}$ such that, for each player $i \in \{1,2\}$, the following holds:*

$$
\mathbb{E}_{x\sim\mathcal{D}}[u_i(x)] \geq \lambda_2 \cdot \max_{x\in\mathcal{P}}(x - c_i) f(x),
$$

*where $\lambda_2 \in (0,1)$ is an absolute constant depending on $\lambda_1$ and independent of $k$.*

*Proof.* Let $\mathcal{P}_i := \{c_i, c_i + 1/k, c_i + 2/k, \ldots, 1\}$. Let $s_{\max}^{(i)} := \max_{j\in\mathcal{P}_i}(j/k - c_i) \cdot f(j/k)$. Let us assume that $s_{\max}^{(2)} \geq \lambda_1 \cdot s_{\max}^{(1)}$.

Let $x_0 \in \mathcal{P}$ be the smallest price such that $c_1 < x_0 < c_2 - 1/k$ and $(x_0 - c_1) \cdot f(x_0) \geq \frac{\lambda_1}{8e^2} s_{\max}^{(1)}$. Observe that such a price exists due to our assumptions and $\lambda_0 := \frac{(x_0 - c_1) \cdot f(x_0)}{s_{\max}^{(1)}} \leq \frac{\lambda_1}{4e^2}$. Now we define the CCE $\mathcal{D}$ as follows. We choose the pair of prices $(x_0, x_0 + 1/k)$ with probability $p_0 := 1 - \lambda_0$.

Let $s_i := (i/k - c_2) \cdot f(i/k)$ and $S_i := \max_{j \leq i} s_j$. Consider the smallest index $m$ such that $m \in \arg\max_{i \in [k]} s_i$. Consider a constant $\lambda := \frac{1}{2e^2}$ and set $B := \lambda s_{\max}^{(2)}$. We now consider the following two cases.

**Case 1:** $s_{k \cdot c_2 + 1} \geq B$. We choose the price $c_2 + 1/k$ for both the players with probability $1 - p_0$. In this case, observe that $\mathbb{E}_{(x_1, x_2) \sim \mathcal{D}}[u_2(x_1, x_2)] = (1 - p_0) s_{k \cdot c_2 + 1}/2 \geq (1 - p_0) B/2$ and $\max_{x \in \mathcal{P}} \mathbb{E}_{(x_1, x_2) \sim \mathcal{D}}[u_2(x_1, x)] = (1 - p_0) s_{k \cdot c_2 + 1}/2$.

**Case 2:** $s_{k \cdot c_2 + 1} < B$. Let $i_0 := \min\{i : S_i \geq B\}$. We randomly choose a price $x = i/k$ for both the players with probability $(1 - p_0)(\tau_i - \tau_{i+1})$, where $\tau_i$ is defined as follows:

$$\tau_i := \begin{cases} 1, & i < i_0, \\ B/S_i, & i_0 \leq i \leq m, \\ 0, & i > m, \end{cases}$$

In this case, using analogous calculations as that of the proof of Theorem 2.1, we can show that the following:

$$\mathbb{E}_{(x_1, x_2) \sim \mathcal{D}}[u_2(x_1, x_2)] \geq (1 - p_0) B.$$

Similarly, using analogous calculations as that of the proof of Theorem 2.1, we can show that the following:

$$\max_{x \in \mathcal{P}} \mathbb{E}_{(x_1, x_2) \sim \mathcal{D}}[u_2(x_1, x)] \leq (1 - p_0) B.$$

Now for either case, we want to show that player 1 will have no incentive to unilaterally deviate under the CCE $\mathcal{D}$. First we have the following:

$$\mathbb{E}_{(x_1, x_2) \sim \mathcal{D}}[u_1(x_1, x_2)] \geq p_0 \cdot \lambda_0 \cdot s_{\max}^{(1)} + \frac{(1 - p_0)}{4e^2} \cdot \lambda_1 \cdot s_{\max}^{(1)}$$

Next we have the following:

$$\max_{x \in \mathcal{P} : x \leq x_0 + 1/k} \mathbb{E}_{(x_1, x_2) \sim \mathcal{D}}[u_1(x, x_2)] = \lambda_0 \cdot s_{\max}^{(1)}$$

Next we have the following:

$$\max_{x \in \mathcal{P} : x > x_0 + 1/k} \mathbb{E}_{(x_1, x_2) \sim \mathcal{D}}[u_1(x, x_2)] \leq (1 - p_0) s_{\max}^{(1)}$$

As $\lambda_0 \leq \lambda_1/(4e^2)$ and $p_0 = 1 - \lambda_0$, then we have the following:

$$\mathbb{E}_{(x_1, x_2) \sim \mathcal{D}}[u_1(x_1, x_2)] \geq \max_{x \in \mathcal{P}} \mathbb{E}_{(x_1, x_2) \sim \mathcal{D}}[u_1(x, x_2)]$$

$\square$

We next prove the following technical lemma for costs $c_1 \leq c_2$.

**Lemma A.3.** *Consider a distribution $\mathcal{D}$ over the prices in $\mathcal{P}$. Let $i_\star/k$ be the largest price that has positive weight under the distribution $\mathcal{D}$, and let $\mathcal{P}_i := \{i_\star/k, (i_\star + 1)/k, \ldots, 1\} \setminus \{1/k, 2/k, \ldots, c_i + 2/k\}$. If $i_\star/k > c_1$ and $f(c_1 + 1/k) > 0$, then we have*

$$\max_{x_1 \in \mathcal{P} \setminus \mathcal{P}_1} \mathbb{E}_{x_2 \sim \mathcal{D}}[u_1(x_1, x_2)] > \max_{x_1 \in \mathcal{P}_1} \mathbb{E}_{x_2 \sim \mathcal{D}}[u_1(x_1, x_2)],$$

*and if $i_\star/k > c_2$ and $f(c_2 + 1/k) > 0$, then we have*

$$\max_{x_2 \in \mathcal{P} \setminus \mathcal{P}_2} \mathbb{E}_{x_1 \sim \mathcal{D}}[u_2(x_1, x_2)] > \max_{x_2 \in \mathcal{P}_2} \mathbb{E}_{x_1 \sim \mathcal{D}}[u_1(x_1, x_2)].$$

*Proof.* Let us assume that $i_\star/k > c_1$. Observe that $\mathbb{E}_{x_2 \sim \mathcal{D}}[u_1(c_1 + 1/k, x_2)] > 0$ and $\mathbb{E}_{x_2 \sim \mathcal{D}}[u_1(i/k, x_2)] = 0$ for all $i > i_\star$. Next we have the following if $i_\star \geq kc_1 + 3$ and $f(\frac{i_\star - 1}{k}) > 0$:

$$\mathbb{E}_{x_2 \sim \mathcal{D}}\left[u_1\left(\frac{i_\star - 1}{k}, x_2\right)\right] \geq \frac{i_\star - 1 - kc_1}{k} \cdot f\left(\frac{i_\star - 1}{k}\right) \cdot \mathbb{P}_{x \sim \mathcal{D}}[x = i_\star/k]$$

$$> \frac{i_\star - kc_1}{2k} \cdot f\left(\frac{i_\star}{k}\right) \cdot \mathbb{P}_{x \sim \mathcal{D}}[x = i_\star/k]$$

$$= \mathbb{E}_{x_2 \sim \mathcal{D}}\left[u_1\left(\frac{i_\star}{k}, x_2\right)\right]$$

If $f\left(\frac{i_\star - 1}{k}\right) = 0$, then $\mathbb{E}_{x_2 \sim \mathcal{D}}[u_1(i_\star/k, x_2)] = 0$ whereas $\mathbb{E}_{x_2 \sim \mathcal{D}}[u_1(c_1 + 1/k, x_2)] > 0$.

Analogously, we can prove that $\max_{x_2 \in \mathcal{P} \setminus \mathcal{P}_2} \mathbb{E}_{x_1 \sim \mathcal{D}}[u_2(x_1, x_2)] > \max_{x_2 \in \mathcal{P}_2} \mathbb{E}_{x_1 \sim \mathcal{D}}[u_1(x_1, x_2)]$ if $i_\star/k > c_2$. $\qquad \square$

*Proof of Theorem 2.7.* As $\mathcal{D}$ is a CE, then we always have $\mathbb{E}_{x \sim \mathcal{D}}[u_2(x)] \geq 0$. If $f(c_2 + 1/k) = 0$, then $\mathbb{E}_{x \sim \mathcal{D}}[u_2(x)] = 0$. Let us assume that $f(c_2 + 1/k) > 0$.

We now prove that for any pair of prices $(x_1, x_2)$ which has a positive weight under the CE $\mathcal{D}$, we have $x_1 \leq c_2 + \frac{2}{k}$. For the sake of contradiction, let us assume that this is not true. Let $i_1/k$ be the largest price that the player 1 chooses under $\mathcal{D}$ and let $i_2/k$ be the largest price that the player 2 chooses under $\mathcal{D}$. First, let us consider the case where $i_1 \geq i_2$. Note that $i_1/k \geq c_2 + 3/k$. Now due to Lemma A.3, we have the following:

$$\mathbb{E}_{(x_1, x_2) \sim \mathcal{D}}[u_1(i_1/k, x_2)|x_1 = i_1/k] < \max_{x \in \mathcal{P}} \mathbb{E}_{(x_1, x_2) \sim \mathcal{D}}[u_1(x, x_2)|x_1 = i_1/k]$$

This is a contradiction to our assumption that $\mathcal{D}$ is a correlated equilibrium. Next, let us consider the case where $i_2 > i_1$. Note that $i_2/k \geq c_2 + 3/k$. Now due to Lemma A.3, we have the following:

$$\mathbb{E}_{(x_1, x_2) \sim \mathcal{D}}[u_2(x_1, i_2/k)|x_2 = i_2/k] < \max_{x \in \mathcal{P}} \mathbb{E}_{(x_1, x_2) \sim \mathcal{D}}[u_1(x_1, x)|x_2 = i_2/k]$$

This is a contradiction to our assumption that $\mathcal{D}$ is a correlated equilibrium.

Hence, for any pair of prices $(x_1, x_2)$ which has a positive weight under the CE $\mathcal{D}$, we have $x_1 \leq c_2 + \frac{2}{k}$. If for any pair of prices $(x_1, x_2)$ which has a positive weight under the CE $\mathcal{D}$ we have $x_1 \leq c_2$ then $\mathbb{E}_{x \sim \mathcal{D}}[u_2(x)] = 0$. If there is a pair $(x_1, x_2)$ which has a positive weight under the CE $\mathcal{D}$ and $x_1 \in \{c_2 + 1/k, c_2 + 2/k\}$, then due to Lemma A.3, we have $x_2 \leq c_2 + 2/k$. Now we claim that $x_2 \in \{c_2 + 1/k, c_2 + 2/k\}$. If this is not the case, player 2 can deviate from $x_2$ to $c_2 + 1/k$ and get a higher reward. Hence, in this case we have $\mathbb{E}_{x \sim \mathcal{D}}[u_2(x)] \leq 1/k$.

Now let us assume that $c_2 \geq c_1 + 3/k$. We claim that for any pair of prices $(x_1, x_2)$ which has a positive weight under the CE $\mathcal{D}$, we have $x_1 \leq c_2$. For the sake of contradiction, let us assume that this is not true. In this case, recall that $x_2 \in \{c_2 + 1/k, c_2 + 2/k\}$. Let us first consider the case where there is a pair $(x_1, x_2)$ which has a positive weight under the CE $\mathcal{D}$ and $x_1 = c_2 + 2/k$. In this case, we have the following:

$$\mathbb{E}_{(x_1, x_2) \sim \mathcal{D}}[u_1(c_2, x_2)|x_1 = c_2 + 2/k] = (c_2 - c_1) \cdot f(c_2)$$

$$> \frac{1}{2} \cdot (c_2 + 2/k - c_1) \cdot f(c_2 + 2/k)$$

$$\geq \mathbb{E}_{(x_1, x_2) \sim \mathcal{D}}[u_1(c_2 + 2/k, x_2)|x_1 = c_2 + 2/k]$$

This is a contradiction to our assumption that $\mathcal{D}$ is a correlated equilibrium. Next let us consider the case where there is a pair $(x_1, x_2)$ which has a positive weight under the CE $\mathcal{D}$ and $x_1 = c_2 + 1/k$. In this case, we have $x_2 = c_2 + 1/k$ as $\mathcal{D}$ is a correlated equilibrium. In this case, we have the following:

$$\mathbb{E}_{(x_1, x_2) \sim \mathcal{D}}[u_1(c_2, x_2)|x_1 = c_2 + 1/k] = (c_2 - c_1) \cdot f(c_2)$$

$$> \frac{1}{2} \cdot (c_2 + 1/k - c_1) \cdot f(c_2 + 1/k)$$

$$= \mathbb{E}_{(x_1, x_2) \sim \mathcal{D}}[u_1(c_2 + 1/k, x_2)|x_1 = c_1 + 2/k]$$

This is a contradiction to our assumption that $\mathcal{D}$ is a correlated equilibrium. Hence, if $c_2 \geq c_1 + 3/k$, then for any pair of prices $(x_1, x_2)$ which has a positive weight under the CE $\mathcal{D}$, we have $x_1 \leq c_2$.

Next let us consider the case $c_2 = c_1 + 2/k$. We claim that for any pair of prices $(x_1, x_2)$ which has a positive weight under the CE $\mathcal{D}$, we have $x_1 \leq c_2$. For the sake of contradiction, let us assume that this is not true. In this case, recall that $x_2 \in \{c_2 + 1/k, c_2 + 2/k\}$. Let us first consider the case where there is a pair $(x_1, x_2)$ which has a positive weight under the CE $\mathcal{D}$ and $x_1 = c_2 + 2/k$. Since $\mathcal{D}$ is a correlated equilibrium, we have zero weight on $(c_2 + 2/k, c_2 + 1/k)$ otherwise player 1 can deviate from $c_2 + 2/k$ to $c_2 + 1/k$ and get a higher reward. Hence, we have the following:

$$
\begin{aligned}
\mathbb{E}_{(x_1,x_2)\sim\mathcal{D}}[u_1(c_2 + 1/k, x_2)|x_1 = c_2 + 2/k] &= (c_2 + 1/k - c_1) \cdot f(c_2 + 1/k) \\
&> \frac{1}{2} \cdot (c_2 + 2/k - c_1) \cdot f(c_2 + 2/k) \\
&\geq \mathbb{E}_{(x_1,x_2)\sim\mathcal{D}}[u_1(c_2 + 2/k, x_2)|x_1 = c_2 + 2/k]
\end{aligned}
$$

This is a contradiction to our assumption that $\mathcal{D}$ is a correlated equilibrium.

Next let us consider the case where there is a pair $(x_1, x_2)$ which has a positive weight under the CE $\mathcal{D}$ and $x_1 = c_2 + 1/k$. In this case, we have $x_2 = c_2 + 1/k$ as $\mathcal{D}$ is a correlated equilibrium. In this case, we have the following:

$$
\begin{aligned}
\mathbb{E}_{(x_1,x_2)\sim\mathcal{D}}[u_1(c_2, x_2)|x_1 = c_2 + 1/k] &= (c_2 - c_1) \cdot f(c_2) \\
&> \frac{1}{2} \cdot (c_2 + 1/k - c_1) \cdot f(c_2 + 1/k) \\
&= \mathbb{E}_{(x_1,x_2)\sim\mathcal{D}}[u_1(c_2 + 1/k, x_2)|x_1 = c_2 + 2/k]
\end{aligned}
$$

This is a contradiction to our assumption that $\mathcal{D}$ is a correlated equilibrium.

Hence, if $c_2 = c_1 + 2/k$, then for any pair of prices $(x_1, x_2)$ which has a positive weight under the CE $\mathcal{D}$, we have $x_1 \leq c_2$. $\qquad \square$

We prove Theorem 2.8 in the following two theorems.

**Theorem A.4.** *For the constant demand function $f(x) = 1$, there exists marginal costs $c_1 < c_2$ and a $\Phi$-correlated equilibrium $\mathcal{D}$ in which $\Phi_1$ contains all deviation maps and $\Phi_2$ contains only constant deviation maps such that, for each player $i \in \{1, 2\}$ and large $k$, the following holds:*

$$
\mathbb{E}_{x\sim\mathcal{D}}[u_i(x)] \geq \lambda_0 \cdot \max_{x\in\mathcal{P}}(x - c_i)\, f(x),
$$

*where $\lambda_0 > 0$ is an absolute constant independent of $k$.*

*Proof.* Let us assume that $k \geq 36e^{10} + 1$. Fix an integer $M := \left\lfloor \frac{k-1}{e^{10}} \right\rfloor$. Let $c_1 = 0$ and $c_2 = \frac{\lfloor M/36 \rfloor}{k}$. Now we describe a $\Phi$-correlated equilibrium $\mathcal{D}$ in which $\Phi_1$ contains all deviation maps and $\Phi_2$ contains only constant deviation maps. We begin by describing the marginal distribution $\mathcal{D}_1$ of the prices chosen by the player 1.

We now define $p_j$, the probability that the player 1 chooses the price $j/k$, as follows:

$$
p_j = \begin{cases}
0 & \text{if } j < M \text{ or } j = k, \\
\frac{1}{M+1} & \text{if } j = M, \\
\frac{M}{j(j+1)} & \text{if } M < j < k - 1, \\
\frac{M}{k-1} & \text{if } j = k - 1.
\end{cases}
$$

The above values are nonnegative and sum to 1 which we prove later.

Next, we describe how player 2 selects a price conditional on the price chosen by player 1. Given that player 1 chooses the price $i/k$, player 2 chooses the price $(i+1)/k$ with probability $1/2$ and the price $\lfloor i/2 \rfloor /k$ with probability $1/2$.

First, we show that the player 1 has low swap regret. Consider a price $i/k$ such that $p_i > 0$. First, observe that $\mathbb{E}_{(x_1,x_2)\sim\mathcal{D}}[u_1(x_1,x_2)|x_1 = i/k] = i/(2k)$. Next observe that $\mathbb{E}_{(x_1,x_2)\sim\mathcal{D}}[u_1((i + 1)/k, x_2)|x_1 = i/k] = (i + 1)/(4k) \leq i/(2k)$. Next observe that $\mathbb{E}_{(x_1,x_2)\sim\mathcal{D}}[u_1(j/k, x_2)|x_1 = i/k] = 0$ for all $j > i + 1$. Next observe that $\mathbb{E}_{(x_1,x_2)\sim\mathcal{D}}[u_1(j/k, x_2)|x_1 = i/k] = j/(2k) < i/(2k)$ for all $\lfloor i/2 \rfloor < j < i$. Finally, observe that $\mathbb{E}_{(x_1,x_2)\sim\mathcal{D}}[u_1(j/k, x_2)|x_1 = i/k] \leq i/(2k)$ for all $j \leq \lfloor i/2 \rfloor$. Hence, the player 1 has low swap regret.

Next, we show that the player 2 has low external regret.

Let $S_i := \sum_{j\geq i} p_j$. We have the following for all $i \in [k-1]$ (which we prove later):

$$S_i = \min\left(1, \frac{M}{i}\right),$$

so

$$\mathbb{E}_{x\sim\mathcal{D}_1}[x] = \frac{1}{k}\sum_{j=1}^{k-1} jp_j = \frac{1}{k}\sum_{j=1}^{k-1}\sum_{i=1}^{j} p_j = \frac{1}{k}\sum_{i=1}^{k-1}\sum_{j=i}^{k-1} p_j = \frac{1}{k}\sum_{i=1}^{k-1} S_i$$

$$= \frac{1}{k}\left(M + M\sum_{i=M+1}^{k-1}\frac{1}{i}\right) = \frac{M}{k}\left(1 + H_{k-1} - H_M\right),$$

where $H_n$ is the $n$-th harmonic number.

Now we have the following for any $i \in [k]$:

$$\mathbb{E}_{x_1\sim\mathcal{D}_1}[u_2(x_1, i/k)] = \left(\frac{i}{k} - c_2\right) \cdot \left(\mathbb{P}_{x_1\sim\mathcal{D}_1}(x_1 > i) + \tfrac{1}{2}\mathbb{P}_{x_1\sim\mathcal{D}_1}(x_1 = i)\right) \leq \frac{i}{k}S_i.$$

Hence,

$$\max_{i\in[k]}\mathbb{E}_{x_1\sim\mathcal{D}_1}[u_2(x_1, i/k)] \leq \frac{1}{k}\max_{i\in[k]} i \cdot S_i = \frac{M}{k}.$$

Due to the properties of Harmonic number, we have $H_{k-1} - H_M \geq \ln\frac{k-1}{M} - \frac{1}{2M}$ (which we prove later). With $M = \lfloor(k-1)/e^{10}\rfloor$, we have $H_{k-1} - H_M \geq 9.5$, so

$$\frac{\mathbb{E}_{x\sim\mathcal{D}_1}[x]}{\max_{i\in[k]}\mathbb{E}_{x_1\sim\mathcal{D}_1}[u_2(x_1, i/k)]} \geq 1 + H_{k-1} - H_M \geq 9.5.$$

Next we have the following:

$$\mathbb{E}_{(x_1,x_2)\sim\mathcal{D}}[u_2(x_1, x_2)] = \frac{1}{2}\cdot\sum_{i=M}^{k-1} p_i \cdot \left(\frac{\lfloor i/2 \rfloor}{k} - c_2\right)$$

$$\geq \frac{1}{2}\cdot\sum_{i=M}^{k-1} p_i \cdot \left(\frac{i}{4k} - \frac{i}{36k}\right) \qquad\text{(as } c_2 \leq M/(36k)\text{)}$$

$$= (1/9)\cdot\mathbb{E}_{x\sim\mathcal{D}_1}[x]$$

$$\geq \max_{i\in[k]}\mathbb{E}_{x_1\sim\mathcal{D}_1}[u_2(x_1, i/k)]$$

Hence, player 2 has a low external regret.

Also observe that $\mathbb{E}_{(x_1,x_2)\sim\mathcal{D}}[u_1(x_1, x_2)] \geq (1/2)\cdot\mathbb{E}_{x\sim\mathcal{D}_1}[x]$. Hence, in order to prove the theorem, it suffices to show that $\mathbb{E}_{x\sim\mathcal{D}_1}[x]$ is large. Towards that we have the following:

$$\mathbb{E}_{x\sim\mathcal{D}_1}[x] = \frac{M}{k}\left(1 + H_{k-1} - H_M\right) \geq \frac{4.74}{e^{10}}.$$

**Omitted calculations**

First, we prove that $\sum_{j=1}^{k} p_j = 1$. Using the identity $\frac{1}{j(j+1)} = \frac{1}{j} - \frac{1}{j+1}$, we have the following:

$$
\begin{aligned}
\sum_{j=1}^{k} p_j &= \frac{1}{M+1} + \sum_{j=M+1}^{k-2} \frac{M}{j(j+1)} + \frac{M}{k-1} \\
&= \frac{1}{M+1} + \sum_{j=M+1}^{k-2} \left( \frac{M}{j} - \frac{M}{j+1} \right) + \frac{M}{k-1} \\
&= \frac{1}{M+1} + \frac{M}{M+1} - \frac{M}{k-1} + \frac{M}{k-1} \\
&= 1
\end{aligned}
$$

Next, we prove that $S_i := \sum_{j \geq i} p_j = \left(1, \frac{M}{i}\right)$ for all $i \in [k-1]$. We divide our analysis into three cases.

Case 1: $i \leq M$

As $p_j = 0$ for all $j < M$, we have $\sum_{j<i} p_j = 0$. Therefore

$$
S_i = \sum_{j \geq i} p_j = 1 = \min(1, M/i).
$$

Case 2: $M + 1 \leq i \leq k - 2$

In this case, we have

$$
S_i = \sum_{j=i}^{k-2} \frac{M}{j(j+1)} + \frac{M}{k-1} = \sum_{j=i}^{k-2} \left( \frac{M}{j} - \frac{M}{j+1} \right) + \frac{M}{k-1} = M \left( \frac{1}{i} - \frac{1}{k-1} \right) + \frac{M}{k-1} = \frac{M}{i} = \min(1, M/i).
$$

Case 3: $i = k - 1$

In this case, we have

$$
S_{k-1} = p_{k-1} = \frac{M}{k-1} = \frac{M}{i} = \min(1, M/i).
$$

By combining all the three cases, we get $S_i = \min(1, M/i)$ for all $i \in [k-1]$.

Finally, we show that $H_{k-1} - H_M \geq \ln \frac{k-1}{M} - \frac{1}{2M}$. For any $n \geq 1$, we have the following:

$$
\gamma + \ln n < H_n < \gamma + \ln n + \frac{1}{2n},
$$

where $\gamma$ is the euler's constant and the inequality follows from Guo & Qi (2011).

Hence, we have the following:

$$
H_{k-1} - H_M \geq \gamma + \ln(k-1) - (\gamma + \ln M + \frac{1}{2M}) = \ln \frac{k-1}{M} - \frac{1}{2M}
$$

$\square$

**Theorem A.5.** *For the constant demand function $f(x) = 1$, there exists marginal costs $c_1 > c_2$ and a $\Phi$-correlated equilibrium $\mathcal{D}$ in which $\Phi_1$ contains all deviation maps and $\Phi_2$ contains only constant deviation maps such that, for each player $i \in \{1, 2\}$ and large $k$, the following holds:*

$$
\mathbb{E}_{x \sim \mathcal{D}}[u_i(x)] \geq \lambda_1 \cdot \max_{x \in \mathcal{P}} (x - c_i) f(x),
$$

*where $\lambda_1 > 0$ is an absolute constant independent of $k$.*

*Proof.* Let us assume that $k \geq 36e^{10} + 1$. Fix an integer $M := \left\lfloor \frac{k-1}{e^{10}} \right\rfloor$. Let $c_1 = \frac{\lfloor M/36 \rfloor}{k}$ and $c_2 = 0$. Now we describe a $\Phi$-correlated equilibrium $\mathcal{D}$ in which $\Phi_1$ contains all deviation maps and $\Phi_2$ contains only constant deviation maps. We begin by describing the marginal distribution $\mathcal{D}_1$ of the prices chosen by the player 1.

We now define $p_j$, the probability that the player 1 chooses the price $j/k$, as follows:

$$
p_j = \begin{cases}
0 & \text{if } j < M \text{ or } j = k, \\
\frac{1}{M+1} & \text{if } j = M, \\
\frac{M}{j(j+1)} & \text{if } M < j < k-1, \\
\frac{M}{k-1} & \text{if } j = k-1.
\end{cases}
$$

The above values are nonnegative and sum to 1 (as proven in the previous theorem).

Next, we describe how player 2 selects a price conditional on the price chosen by player 1. Given that player 1 chooses the price $i/k$, player 2 chooses the price $(i+1)/k$ with probability $1/2$ and the price $\lfloor i/2 \rfloor /k$ with probability $1/2$.

First, we show that the player 1 has low swap regret. Consider a price $i/k$ such that $p_i > 0$. First, observe that $\mathbb{E}_{(x_1,x_2)\sim\mathcal{D}}[u_1(x_1,x_2)|x_1 = i/k] = \frac{1}{2}(i/k - c_1) \geq \frac{1}{2}(i/k - i/(36k)) = \frac{35i}{72k}$. Next observe that $\mathbb{E}_{(x_1,x_2)\sim\mathcal{D}}[u_1((i+1)/k,x_2)|x_1 = i/k] = \frac{1}{4} \cdot ((i+1)/k - c_1) \leq \frac{1}{2}(i/k - c_1)$. Next observe that $\mathbb{E}_{(x_1,x_2)\sim\mathcal{D}}[u_1(j/k,x_2)|x_1 = i/k] = 0$ for all $j > i+1$. Next observe that $\mathbb{E}_{(x_1,x_2)\sim\mathcal{D}}[u_1(j/k,x_2)|x_1 = i/k] = \frac{1}{2}(j/k - c_1) < \frac{1}{2}(i/k - c_1)$ for all $\lfloor i/2 \rfloor < j < i$. Finally, observe that $\mathbb{E}_{(x_1,x_2)\sim\mathcal{D}}[u_1(j/k,x_2)|x_1 = i/k] \leq i/(2k) - c_1$ for all $j \leq \lfloor i/2 \rfloor$. Hence, the player 1 has low swap regret.

Next, we show that the player 2 has low external regret.

Let $S_i := \sum_{j \geq i} p_j$. We have the following for all $i \in [k-1]$ (as proven in the previous theorem):

$$
S_i = \min\left(1, \frac{M}{i}\right),
$$

so

$$
\mathbb{E}_{x\sim\mathcal{D}_1}[x] = \frac{1}{k}\sum_{j=1}^{k-1} jp_j = \frac{1}{k}\sum_{j=1}^{k-1}\sum_{i=1}^{j} p_j = \frac{1}{k}\sum_{i=1}^{k-1}\sum_{j=i}^{k-1} p_j = \frac{1}{k}\sum_{i=1}^{k-1} S_i
$$

$$
= \frac{1}{k}\left(M + M\sum_{i=M+1}^{k-1}\frac{1}{i}\right) = \frac{M}{k}\left(1 + H_{k-1} - H_M\right),
$$

where $H_n$ is the $n$-th harmonic number.

Now we have the following for any $i \in [k]$:

$$
\mathbb{E}_{x_1\sim\mathcal{D}_1}[u_2(x_1, i/k)] = \frac{i}{k} \cdot \left(\mathbb{P}_{x_1\sim\mathcal{D}_1}(x_1 > i) + \tfrac{1}{2}\mathbb{P}_{x_1\sim\mathcal{D}_1}(x_1 = i)\right) \leq \frac{i}{k}S_i.
$$

Hence,

$$
\max_{i\in[k]}\mathbb{E}_{x_1\sim\mathcal{D}_1}[u_2(x_1, i/k)] \leq \frac{1}{k}\max_{i\in[k]} i \cdot S_i = \frac{M}{k}.
$$

Due to the properties of Harmonic number, we have $H_{k-1} - H_M \geq \ln\frac{k-1}{M} - \frac{1}{2M}$. With $M = \lfloor (k-1)/e^{10} \rfloor$, we have $H_{k-1} - H_M \geq 9.5$, so

$$
\frac{\mathbb{E}_{x\sim\mathcal{D}_1}[x]}{\max_{i\in[k]}\mathbb{E}_{x_1\sim\mathcal{D}_1}[u_2(x_1, i/k)]} \geq 1 + H_{k-1} - H_M \geq 9.5.
$$

Next we have the following:

$$\mathbb{E}_{(x_1,x_2)\sim\mathcal{D}}[u_2(x_1,x_2)] = \frac{1}{2}\cdot\sum_{i=M}^{k-1} p_i\cdot\frac{\lfloor i/2\rfloor}{k}$$

$$\geq \frac{1}{2}\cdot\sum_{i=M}^{k-1} p_i\cdot\frac{i}{4k}$$

$$= (1/8)\cdot\mathbb{E}_{x\sim\mathcal{D}_1}[x]$$

$$\geq \max_{i\in[k]}\mathbb{E}_{x_1\sim\mathcal{D}_1}[u_2(x_1,i/k)]$$

Hence, player 2 has a low external regret.

Also observe that $\mathbb{E}_{(x_1,x_2)\sim\mathcal{D}}[u_1(x_1,x_2)] \geq (35/72)\cdot\mathbb{E}_{x\sim\mathcal{D}_1}[x]$. Hence, in order to prove the theorem, it suffices to show that $\mathbb{E}_{x\sim\mathcal{D}_1}[x]$ is large. Towards that we have the following:

$$\mathbb{E}_{x\sim\mathcal{D}_1}[x] = \frac{M}{k}\Big(1 + H_{k-1} - H_M\Big) \geq \frac{4.74}{e^{10}}.$$

$\square$

# B. Additional Experiments and Plots

In Appendices B.1, B.2, B.3, and B.4, we provide numerical results for various demand functions and cost levels, following the experimental setup in Section 3. In Appendix B.5, we provide additional empirical results for the Hedge-based no-swap-regret learner. In Appendix B.6, we provide the full set of empirical results for regret matching, using an experimental setup similar to that of the Hedge-based no-swap-regret learner. Finally, in Appendix B.7, we provide intuition for the experimental results in Appendices B.5 and B.6.

## B.1. Numerical experiments for constant demand

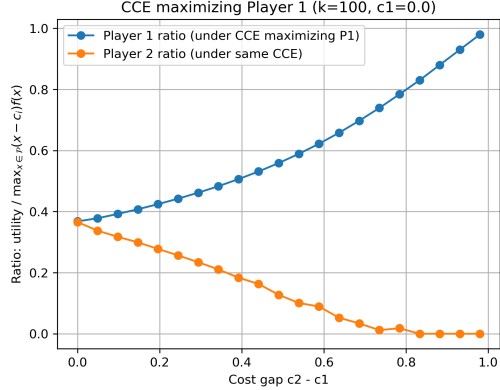
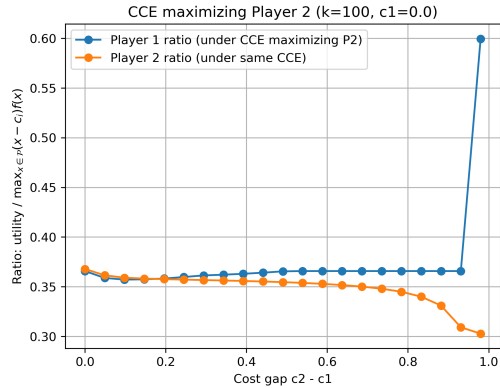

*(a)* Utility Ratios under the CCE that maximizes player 1's utility

*(b)* Utility Ratios under the CCE that maximizes player 2's utility

*Figure 3.* Numerical experiments for asymmetric CCE and constant demand.

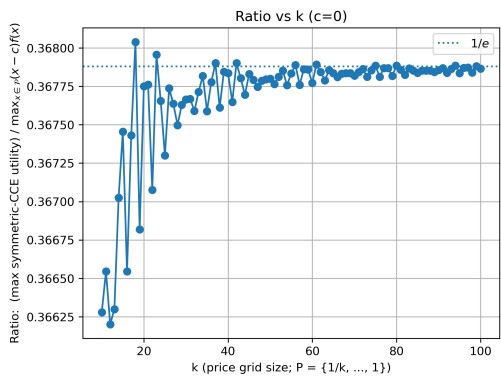

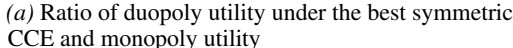

*(a)* Ratio of duopoly utility under the best symmetric CCE and monopoly utility

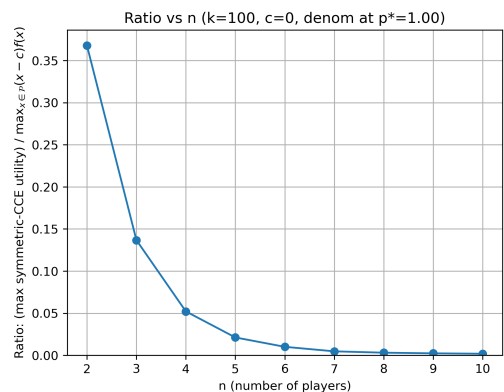

*(b)* Exponential decay of utility under the best symmetric CCE

*Figure 4.* Numerical experiments for symmetric CCE, constant demand and $c = 0.0$.

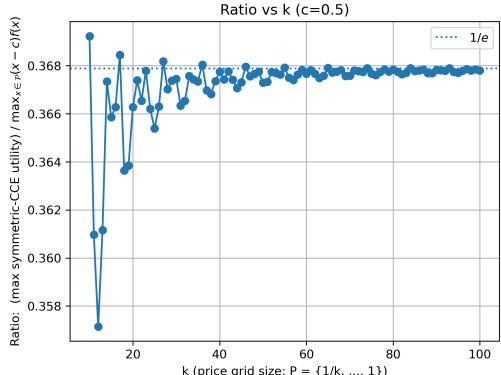

*(a)* Ratio of duopoly utility under the best symmetric CCE and monopoly utility

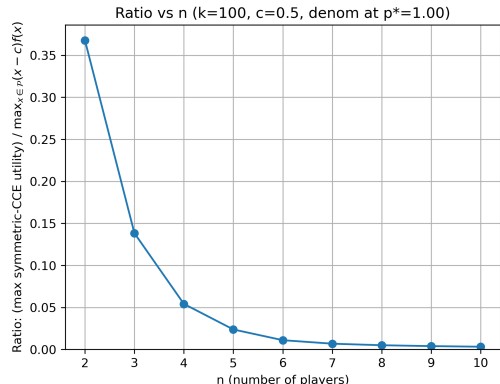

*(b)* Exponential decay of utility under the best symmetric CCE

*Figure 5.* Numerical experiments for symmetric CCE, constant demand and $c = 0.5$.

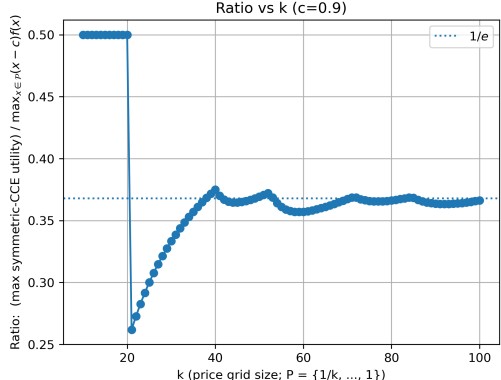

*(a)* Ratio of duopoly utility under the best symmetric CCE and monopoly utility

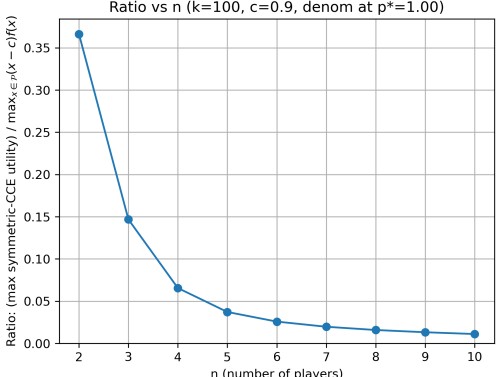

*(b)* Exponential decay of utility under the best symmetric CCE

*Figure 6.* Numerical experiments for symmetric CCE, constant demand and $c = 0.9$.

## B.2. Numerical experiments for linear demand

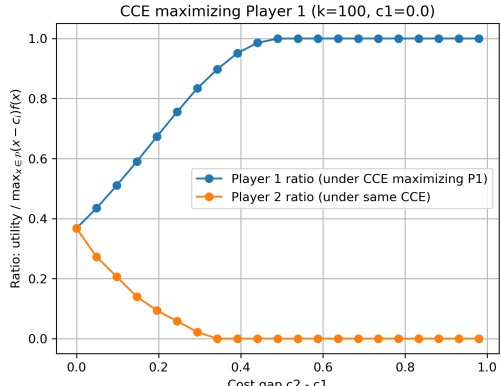

*(a)* Utility Ratios under the CCE that maximizes player 1's utility

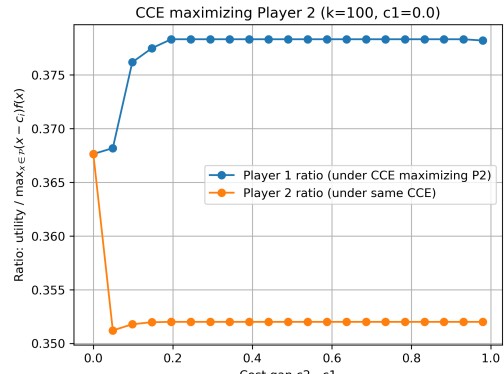

*(b)* Utility Ratios under the CCE that maximizes player 2's utility

*Figure 7.* Numerical experiments for asymmetric CCE and linear demand.

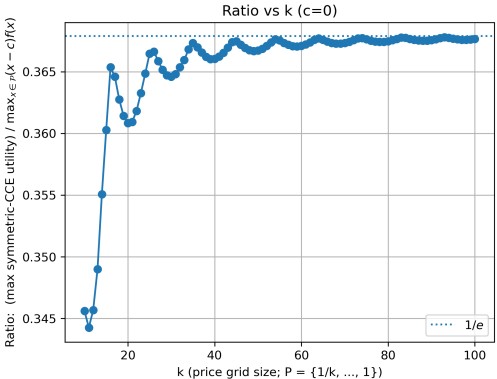

*(a)* Ratio of duopoly utility under the best symmetric CCE and monopoly utility

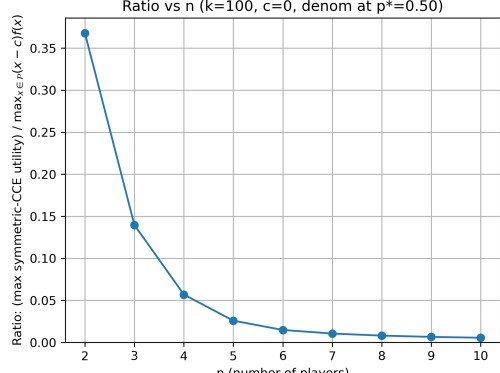

*(b)* Exponential decay of utility under the best symmetric CCE

*Figure 8.* Numerical experiments for symmetric CCE, linear demand and $c = 0.0$.

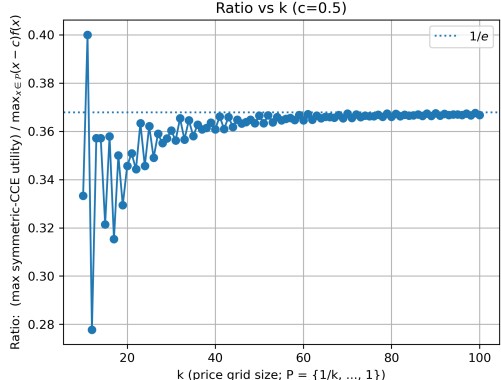

*(a)* Ratio of duopoly utility under the best symmetric CCE and monopoly utility

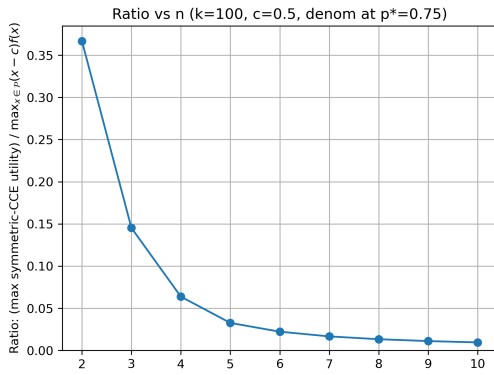

*(b)* Exponential decay of utility under the best symmetric CCE

*Figure 9.* Numerical experiments for symmetric CCE, linear demand and $c = 0.5$.

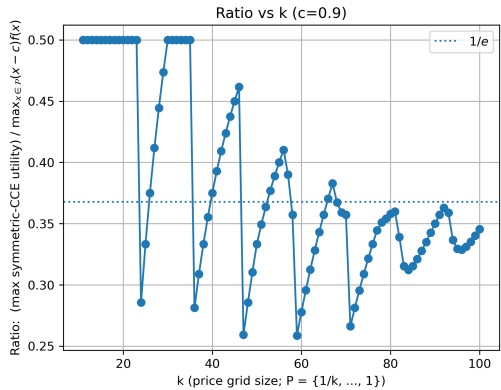
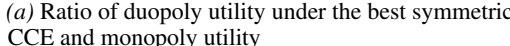

*(a)* Ratio of duopoly utility under the best symmetric CCE and monopoly utility

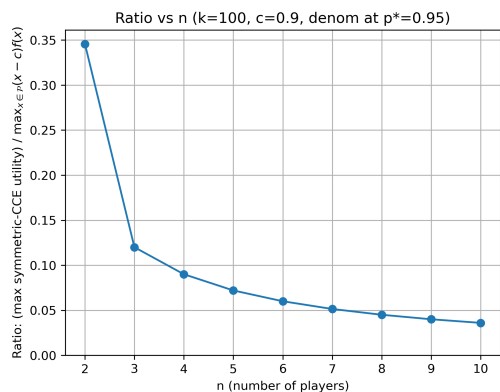

*(b)* Exponential decay of utility under the best symmetric CCE

*Figure 10.* Numerical experiments for symmetric CCE, linear demand and $c = 0.9$.

## B.3. Numerical experiments for quadratic demand

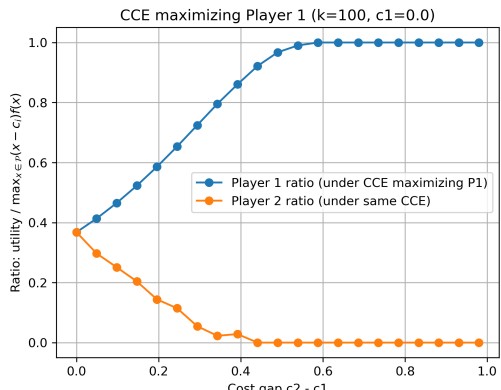

*(a)* Utility Ratios under the CCE that maximizes player 1's utility

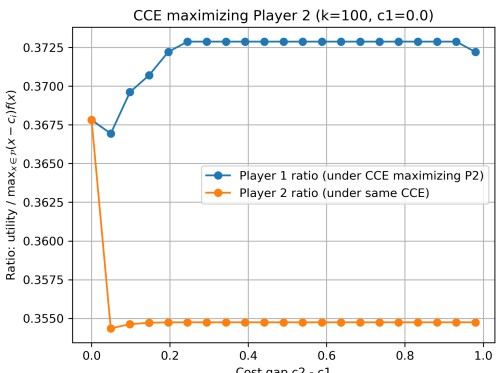

*(b)* Utility Ratios under the CCE that maximizes player 2's utility

*Figure 11.* Numerical experiments for asymmetric CCE and quadratic demand.

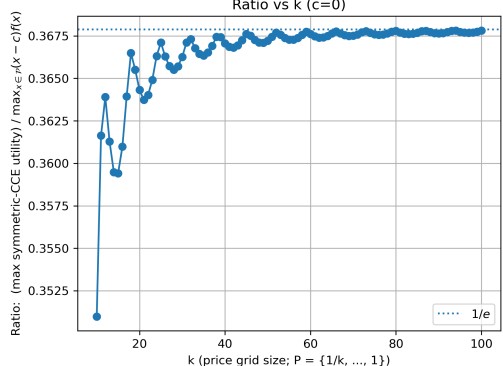

*(a)* Ratio of duopoly utility under the best symmetric CCE and monopoly utility

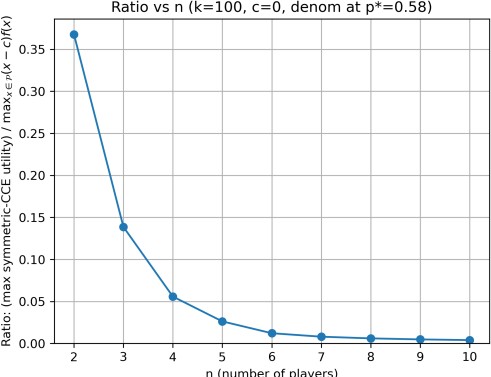

*(b)* Exponential decay of utility under the best symmetric CCE

*Figure 12.* Numerical experiments for symmetric CCE, quadratic demand and $c = 0.0$.

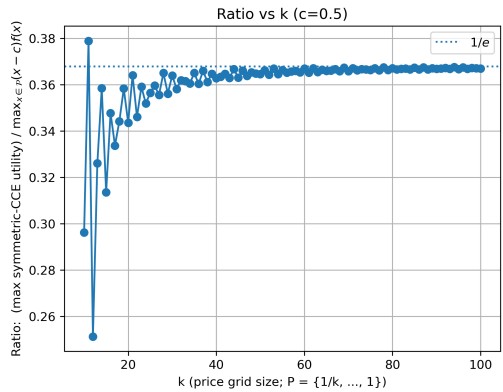
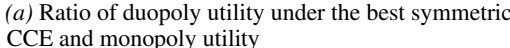

*(a)* Ratio of duopoly utility under the best symmetric CCE and monopoly utility

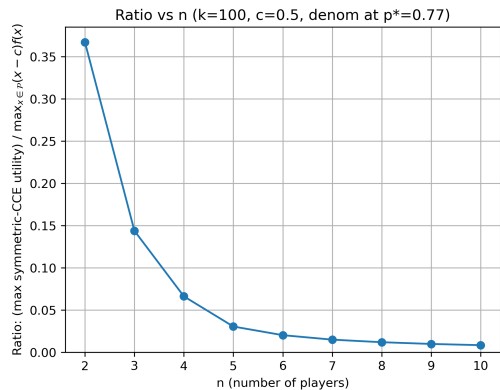

*(b)* Exponential decay of utility under the best symmetric CCE

*Figure 13.* Numerical experiments for symmetric CCE, quadratic demand and $c = 0.5$.

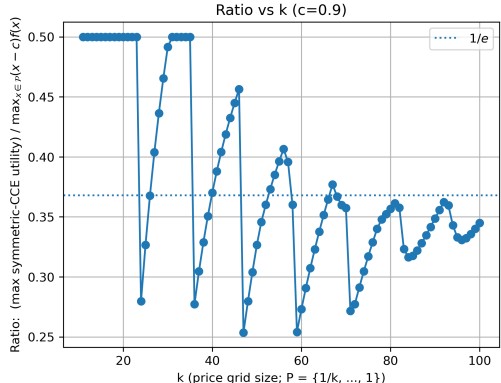

*(a)* Ratio of duopoly utility under the best symmetric CCE and monopoly utility

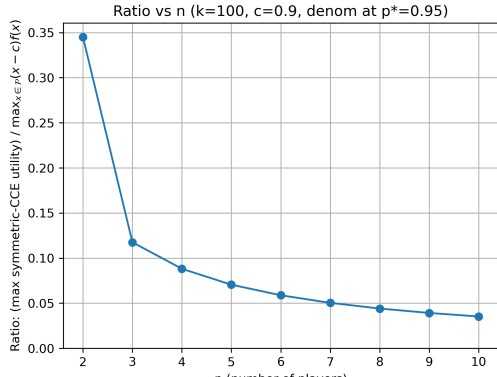

*(b)* Exponential decay of utility under the best symmetric CCE

*Figure 14.* Numerical experiments for symmetric CCE, quadratic demand and $c = 0.9$.

## B.4. Numerical experiments for exponential demand

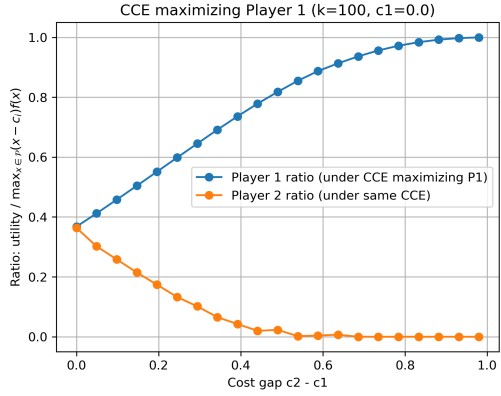

*(a)* Utility Ratios under the CCE that maximizes player 1's utility

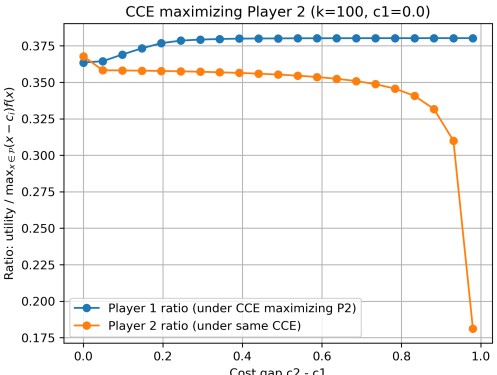

*(b)* Utility Ratios under the CCE that maximizes player 2's utility

*Figure 15.* Numerical experiments for asymmetric CCE and exponential demand.

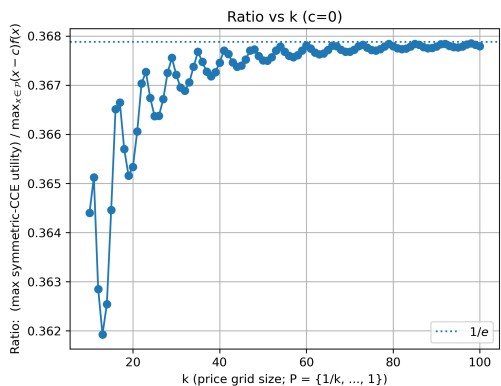

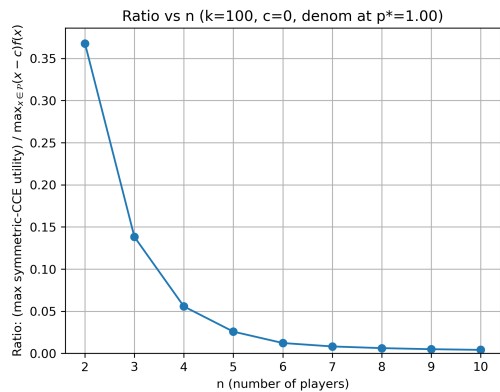

*(a)* Ratio of duopoly utility under the best symmetric CCE and monopoly utility

*(b)* Exponential decay of utility under the best symmetric CCE

*Figure 16.* Numerical experiments for symmetric CCE, exponential demand and $c = 0.0$.

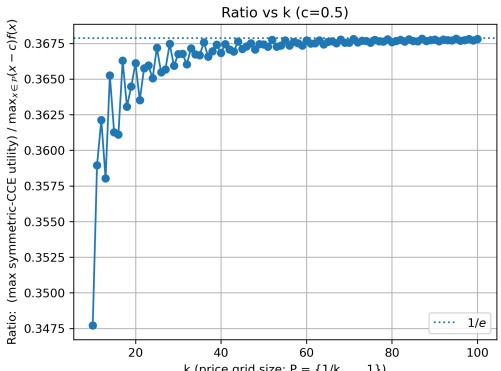

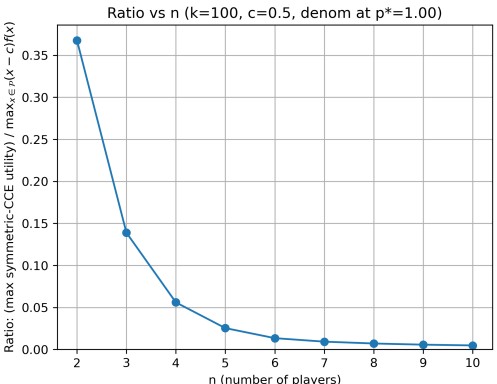

*(a)* Ratio of duopoly utility under the best symmetric CCE and monopoly utility

*(b)* Exponential decay of utility under the best symmetric CCE

*Figure 17.* Numerical experiments for symmetric CCE, exponential demand and $c = 0.5$.

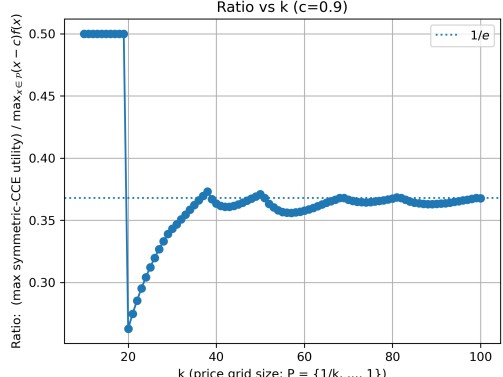

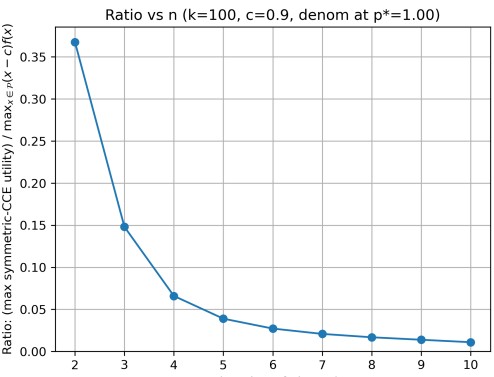

*(a)* Ratio of duopoly utility under the best symmetric CCE and monopoly utility

*(b)* Exponential decay of utility under the best symmetric CCE

*Figure 18.* Numerical experiments for symmetric CCE, quadratic demand and $c = 0.9$.

### B.5. Empirical experiments for Hedge-based no-swap-regret learner

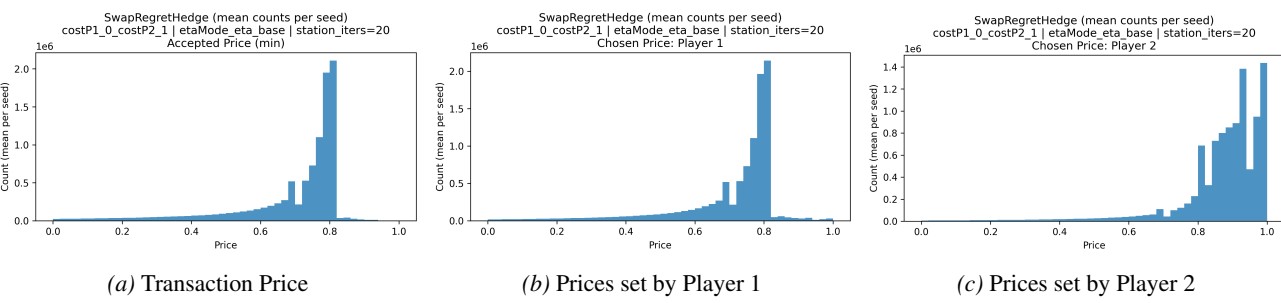

*(a)* Transaction Price  *(b)* Prices set by Player 1  *(c)* Prices set by Player 2

*Figure 19.* Experiments using the Hedge-based no-swap-regret learner with $c_2 = 1$ and both players have the same learning rates. The plots show the frequency of prices over $T = 10^7$ rounds, averaged across 100 random seeds.

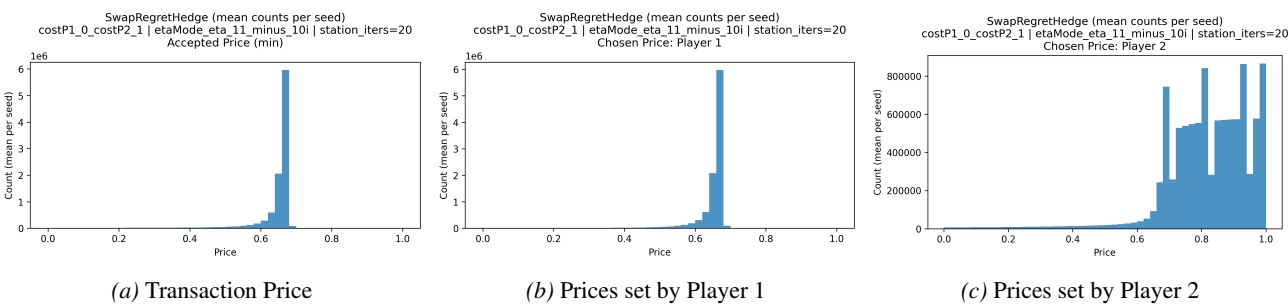

*(a)* Transaction Price  *(b)* Prices set by Player 1  *(c)* Prices set by Player 2

*Figure 20.* Experiments using the Hedge-based no-swap-regret learner with $c_2 = 1$ and player 1 having larger learning rate than player 2. The plots show the frequency of prices over $T = 10^7$ rounds, averaged across 100 random seeds.

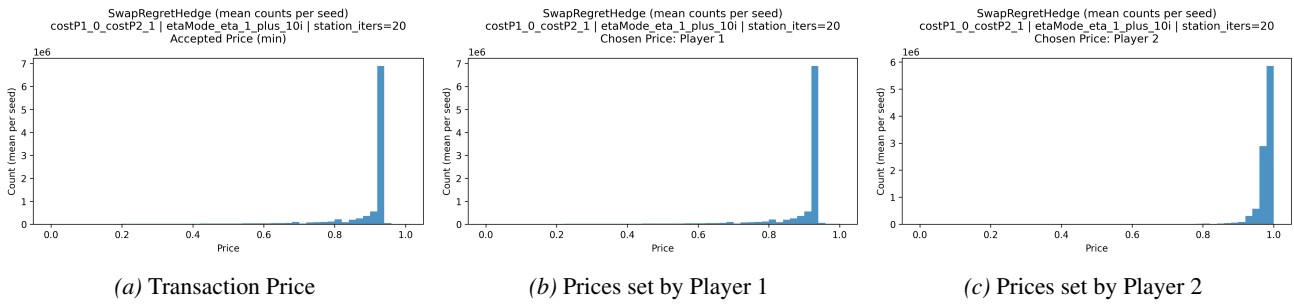

*(a)* Transaction Price  *(b)* Prices set by Player 1  *(c)* Prices set by Player 2

*Figure 21.* Experiments using the Hedge-based no-swap-regret learner with $c_2 = 1$ and player 2 having larger learning rate than player 1. The plots show the frequency of prices over $T = 10^7$ rounds, averaged across 100 random seeds.

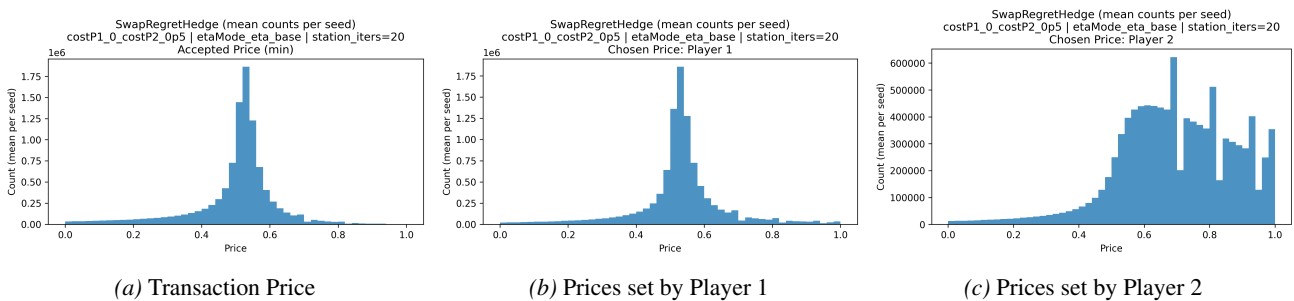

*(a)* Transaction Price  *(b)* Prices set by Player 1  *(c)* Prices set by Player 2

*Figure 22.* Experiments using the Hedge-based no-swap-regret learner with $c_2 = 0.5$ and both players have the same learning rates. The plots show the frequency of prices over $T = 10^7$ rounds, averaged across 100 random seeds.

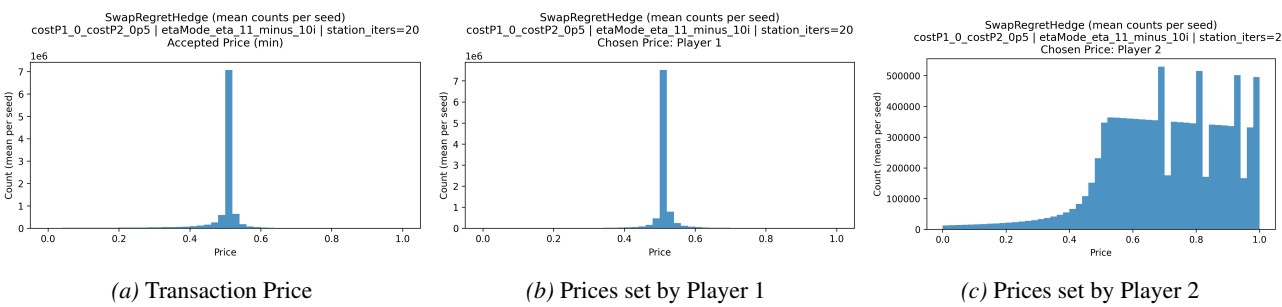

*(a)* Transaction Price        *(b)* Prices set by Player 1        *(c)* Prices set by Player 2

*Figure 23.* Experiments using the Hedge-based no-swap-regret learner with $c_2 = 0.5$ and player 1 having larger learning rate than player 2. The plots show the frequency of prices over $T = 10^7$ rounds, averaged across 100 random seeds.

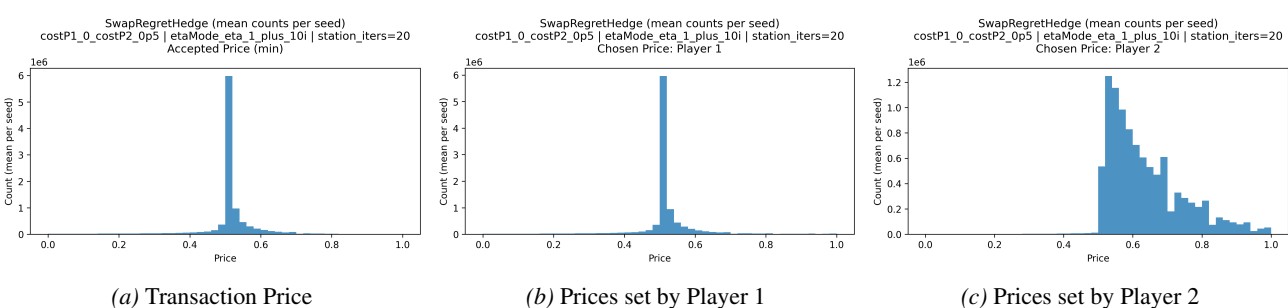

*(a)* Transaction Price        *(b)* Prices set by Player 1        *(c)* Prices set by Player 2

*Figure 24.* Experiments using the Hedge-based no-swap-regret learner with $c_2 = 0.5$ and player 2 having larger learning rate than player 1. The plots show the frequency of prices over $T = 10^7$ rounds, averaged across 100 random seeds.

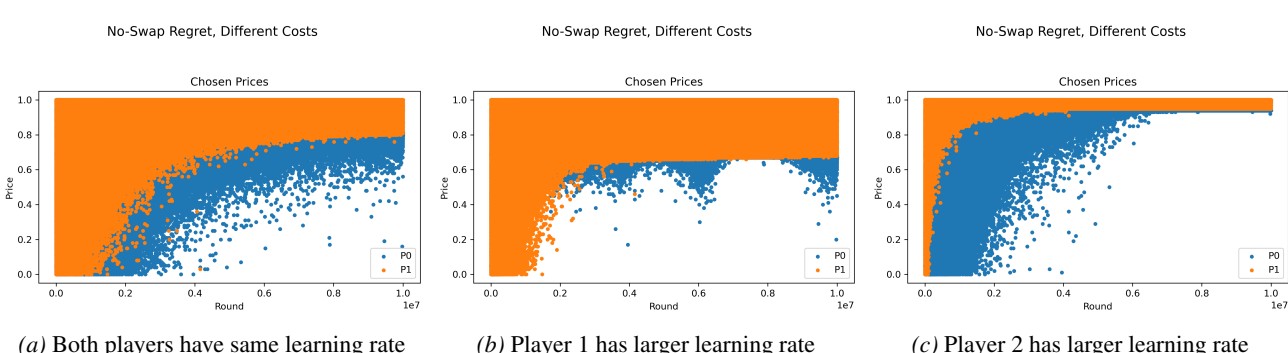

*(a)* Both players have same learning rate     *(b)* Player 1 has larger learning rate     *(c)* Player 2 has larger learning rate

*Figure 25.* Actual prices chosen by each player during $T = 10^7$ rounds for a fixed random seed with $c_2 = 1$. P0 denotes player 1 and P1 denotes player 2.

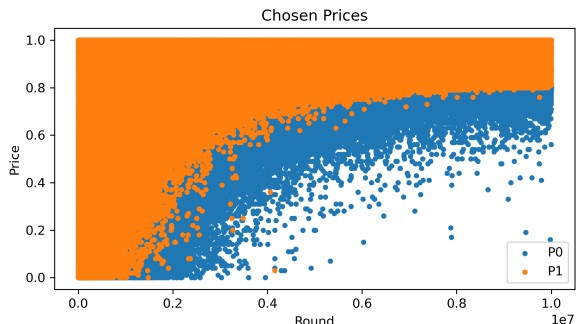 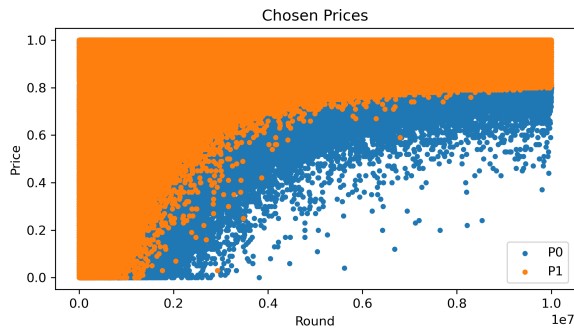

*Figure 26.* Actual prices chosen by each player during $T = 10^7$ rounds for two different random seeds with $c_2 = 1$ and same learning rates. P0 denotes player 1 and P1 denotes player 2. The Hedge-based no-swap regret learner appears to choose the same set of prices across different seeds.

## B.6. Empirical experiments for Regret Matching

In this section, we focus on regret matching. Each player selects prices using a regret-matching algorithm. Whenever the algorithm recommends a price $p$, the player plays $p$ for $t_0$ consecutive rounds and updates the regret-matching algorithm only after these $t_0$ rounds, where $t_0 \in \{1, 20\}$. The plots in this section primarily show the frequency of the prices chosen, while the last two figures depict the actual prices selected in each round under fixed random seeds.

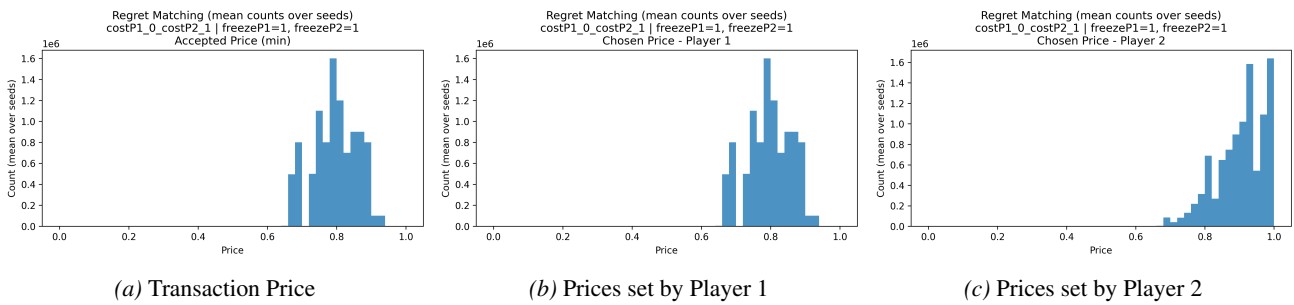

*(a)* Transaction Price        *(b)* Prices set by Player 1        *(c)* Prices set by Player 2

*Figure 27.* Experiments using Regret Matching with $c_2 = 1$ and both players repeating their prices once. The plots show the frequency of prices over $T = 10^7$ rounds, averaged across 100 random seeds.

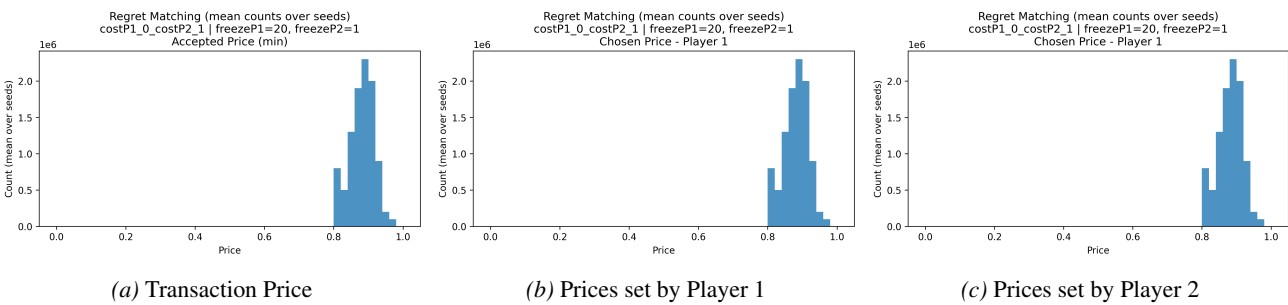

*(a)* Transaction Price        *(b)* Prices set by Player 1        *(c)* Prices set by Player 2

*Figure 28.* Experiments using Regret Matching with $c_2 = 1$ where player 1 repeats its price 20 times and player 2 repeats its price once. The plots show the frequency of prices over $T = 10^7$ rounds, averaged across 100 random seeds.

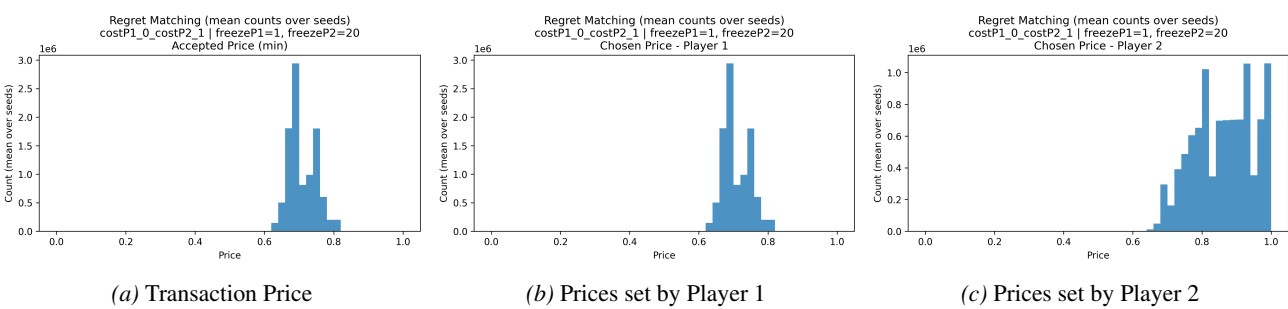

*(a)* Transaction Price      *(b)* Prices set by Player 1      *(c)* Prices set by Player 2

*Figure 29.* Experiments using Regret Matching with $c_2 = 1$ where player 2 repeats its price 20 times and player 1 repeats its price once. The plots show the frequency of prices over $T = 10^7$ rounds, averaged across 100 random seeds.

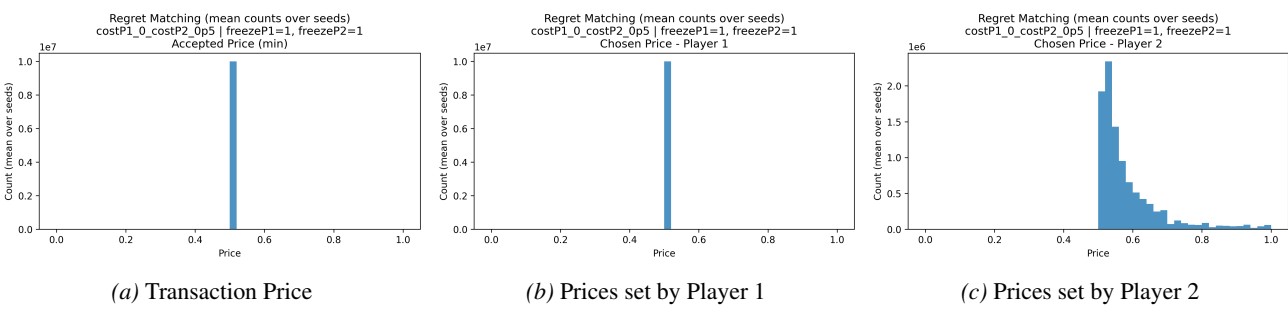

*(a)* Transaction Price      *(b)* Prices set by Player 1      *(c)* Prices set by Player 2

*Figure 30.* Experiments using Regret Matching with $c_2 = 0.5$ and both players repeating their prices once. The plots show the frequency of prices over $T = 10^7$ rounds, averaged across 100 random seeds.

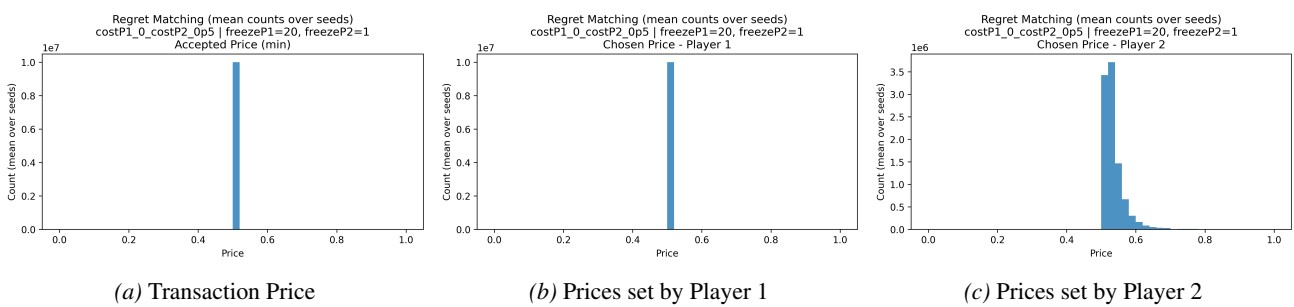

*(a)* Transaction Price      *(b)* Prices set by Player 1      *(c)* Prices set by Player 2

*Figure 31.* Experiments using Regret Matching with $c_2 = 0.5$ where player 1 repeats its price 20 times and player 2 repeats its price once. The plots show the frequency of prices over $T = 10^7$ rounds, averaged across 100 random seeds.

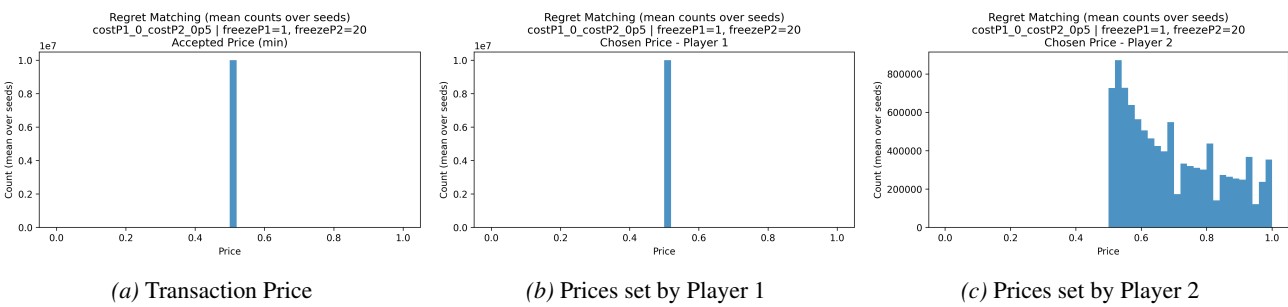

*(a)* Transaction Price      *(b)* Prices set by Player 1      *(c)* Prices set by Player 2

*Figure 32.* Experiments using Regret Matching with $c_2 = 0.5$ where player 2 repeats its price 20 times and player 1 repeats its price once. The plots show the frequency of prices over $T = 10^7$ rounds, averaged across 100 random seeds.

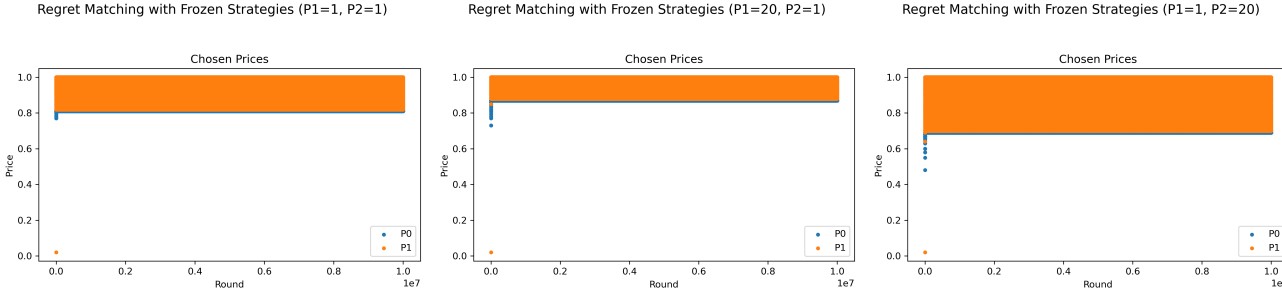

*(a)* Both players repeat their prices once.   *(b)* Player 1 repeats its price 20 times and Player 2 repeats its price once.   *(c)* Player 2 repeats its price 20 times and Player 1 repeats its price once.

*Figure 33.* Actual prices chosen by each player during $T = 10^7$ rounds for a fixed random seed with $c_2 = 1$. P0 denotes player 1 and P1 denotes player 2.

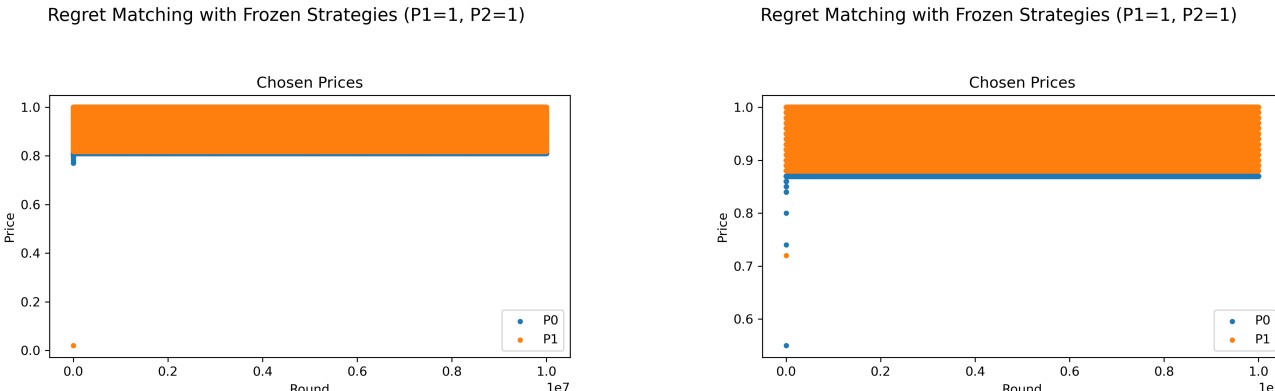

*Figure 34.* Actual prices chosen by each player during $T = 10^7$ rounds for two different random seeds with $c_2 = 1$ and both players repeat their prices once. P0 denotes player 1 and P1 denotes player 2. Regret Matching appears to converge to different prices across different seeds.

### B.7. Intuitive explaination of the plots

We provide intuition for the empirical results in Appendices B.5 and B.6. First, in Appendix B.5, for the case $c_2 = 1$, we observe in Figure 19a that the most frequent transaction price is around $0.8$. When we increase the learning rate of player 1, Figure 20a shows that the most frequent transaction price decreases. Intuitively, a higher learning rate makes player 1 react more aggressively to player 2's posted prices and undercut earlier, which pushes transaction prices down to around $0.7$. In contrast, when we increase the learning rate of player 2, Figure 21a shows that the most frequent transaction price increases to around $0.9$. Intuitively, a higher learning rate makes player 2 respond more quickly to player 1's prices by moving to higher prices sooner, which helps avoid negative utilities and pushes transaction prices upward.

Next, in Appendix B.6, for the case $c_2 = 1$, Figure 27a shows that the most frequent transaction price is around $0.8$. When player 1 repeats its chosen price for 20 consecutive rounds, Figure 28a shows that the most frequent transaction price increases to around $0.9$. Intuitively, repeating prices slows down player 1's response to player 2's posted prices, so player 1 undercuts later than it would without repetition, which pushes transaction prices upward. On the other hand, when player 2 repeats its chosen price for 20 consecutive rounds, Figure 29a shows that the most frequent transaction price decreases to around $0.7$. Intuitively, repetition makes player 2 react more slowly to player 1's posted prices and delay moving to higher prices, which allows player 1 to undercut for longer and pushes transaction prices downward.

Next, observe that in Figure 26, Hedge-based no-swap-regret learners exhibit similar price trajectories across two different random seeds, whereas regret matching produces noticeably different trajectories across seeds. This behavior can be partly explained by the fact that the Hedge-based learner is relatively stable early on due to its learning rate being on the order of

$1/\sqrt{T}$, while regret matching is much more sensitive to the initial random fluctuations, which can in turn influence where transaction prices eventually settle.

Finally, observe that when $c_2 = 0.5$, transaction prices concentrate around $0.5$, and this behavior persists even when we vary the behavior of our no-swap-regret learners. The fact that prices above $0.5$ cannot be sustained in any CE already explains why prices do not rise above this level. To understand why prices also do not fall significantly below $0.5$, consider the limiting regime $T \to \infty$. If the transaction price converges to some level $p$, then player 2 would be playing uniformly at random over prices above $p$. One can then carry out a straightforward calculation showing that no transaction price way below $0.5$ can be sustained in this case. In contrast, when $c_2 = 1$, prices as low as roughly $2/3$ can be sustained in principle.

## C. Additional Technical Details

### C.1. Existence of no-regret learners that converge to a specific equilibrium

In this section, we show that in the duopoly setting where $n = 2$, for every $\Phi$-correlated equilibrium there exist *no-$\Phi_i$-regret* learners for each player $i$ such that, when both players simultaneously use their corresponding learners, the empirical distribution of $\{x^{(t)}\}_{t=1}^T$ converges to that equilibrium as $T \to \infty$.

Fix a $\Phi$-correlated equilibrium $\mu$ on $\mathcal{P}^2$. For every tuple $x \in \mathcal{P}^2$, let $\mu(x)$ denote the weight of $x$ under $\mu$. Fix a time horizon $T$ which is a perfect square. For each tuple $x \in \mathcal{P}^2$, define $n(x) := \lfloor \mu(x) \sqrt{T} \rfloor$ and $T_0 := \sum_{x \in \mathcal{P}^2} n(x)$. Observe that $\sqrt{T} - k^2 \leq T_0 \leq \sqrt{T}$.

Construct a sequence of length $T_0$ by listing each $x$ consecutively $n(x)$ times (and omitting those with $n(x) = 0$). Denote this sequence by $(\hat{x}^t)_{t=1}^{T_0} = ((\hat{x}_1^t, \hat{x}_2^t))_{t=1}^{T_0}$.

Define the empirical distribution of this $T_0$-length sequence:

$$\tilde{\mu}(x) := \frac{1}{T_0} \left| \{t \in [T_0] : \hat{x}^t = x\} \right| = \frac{n(x)}{T_0}.$$

Now we claim that $\|\tilde{\mu} - \mu\|_1 \leq \frac{2k^2}{\sqrt{T}}$. First we have the following:

$$\|\tilde{\mu} - \mu\|_1 \leq \sum_x \left| \frac{n(x)}{T_0} - \frac{n(x)}{\sqrt{T}} \right| + \sum_x \left| \frac{n(x)}{\sqrt{T}} - \mu(x) \right|.$$

The second sum is at most $k^2/\sqrt{T}$ as $\left| \frac{n(x)}{\sqrt{T}} - \mu(x) \right| \leq 1/\sqrt{T}$ for each $x \in \mathcal{P}^2$. For the first sum, we have the following:

$$\sum_x \left| \frac{n(x)}{T_0} - \frac{n(x)}{\sqrt{T}} \right| = \left| \frac{1}{T_0} - \frac{1}{\sqrt{T}} \right| \sum_x n(x) = \left| \frac{1}{T_0} - \frac{1}{\sqrt{T}} \right| T_0 = \left| 1 - \frac{T_0}{\sqrt{T}} \right| = \frac{\sqrt{T} - T_0}{\sqrt{T}} \leq \frac{k^2}{\sqrt{T}}.$$

Hence, $\|\tilde{\mu} - \mu\|_1 \leq \frac{2k^2}{\sqrt{T}}$.

Fix player $i \in \{1, 2\}$. Let $\mathcal{A}_i$ be any standard no-$\Phi_i$-regret algorithm. Define learner $\mathcal{L}_i$ that operates epoch-by-epoch, with each epoch lasting $T_0$ rounds (except possibly the last partial epoch). Let $x^{s,t}$ denote the price tuple played in round $t \in [T_0]$ of epoch $s$. Starting at epoch $s = 1$ and round $t = 1$, player $i$ plays the prescribed action $x_i^{s,t} = \hat{x}_i^t$. If in any round $t$ of epoch $s$ it observes $x_{-i}^{s,t} \neq \hat{x}_{-i}^t$, then from the next round onward it switches permanently to $\mathcal{A}_i$.

If each player $i$ uses the learner $\mathcal{L}_i$, then no mismatch ever occurs, and the price tuples chosen is the periodic repetition of the $T_0$-length cycle $(\hat{x}^t)_{t=1}^{T_0}$, up to at most one partial epoch of length at most $T_0$. Hence we have the following:

$$\left\| \frac{1}{T} \sum_{t=1}^T \mathbf{1}\{x^{(t)} = \cdot\} - \tilde{\mu} \right\|_1 \leq \frac{T_0}{T}.$$

Since $T_0 \leq \sqrt{T}$, we have $T_0/T \to 0$ as $T \to \infty$. Also recall that $\|\tilde{\mu} - \mu\|_1 \to 0$ as $T \to \infty$, so the empirical distribution of $\{x^{(t)}\}_{t=1}^T$ converges to $\mu$.

We now compute the $\Phi_i$-regret of player $i$.

Let us first consider the case where no mismatch is observed. Fix $\phi \in \Phi_i$ and define the one-shot deviation gain

$$\Delta_i^\phi(x) := u_i(\phi(x_i), x_{-i}) - u_i(x).$$

As $\mu$ is a $\Phi$-correlated equilibrium, $\mathbb{E}_{x \sim \mu}[\Delta_i^\phi(x)] \leq 0$. Also the utilities of player $i$ lies in $[-c_i, 1 - c_i]$ in this game, so $|\Delta_i^\phi(x)| \leq 1$ for all $x \in \mathcal{P}^2$. Therefore

$$\mathbb{E}_{x \sim \tilde{\mu}}[\Delta_i^\phi(x)] \leq \mathbb{E}_{x \sim \mu}[\Delta_i^\phi(x)] + \|\tilde{\mu} - \mu\|_1 \leq \|\tilde{\mu} - \mu\|_1.$$

As no mismatch is observed, the sequence of price tuples played is a repetition of the cycle whose empirical distribution is exactly $\tilde{\mu}$ on each full epoch, plus a final partial epoch of length $< T_0$. Hence

$$\sum_{t=1}^T \Delta_i^\phi(x^{(t)}) \leq T \cdot \|\tilde{\mu} - \mu\|_1 + T_0 \leq T \cdot \frac{2k^2}{\sqrt{T}} + \sqrt{T} = O(k^2\sqrt{T}).$$

Let us now consider a case where a mismatch is observed. Let $\tau$ be the first time-step a mismatch is detected. Recall that from time-step $\tau + 1$ onward, player $i$ runs $\mathcal{A}_i$, whose $\Phi_i$-regret over the remaining rounds is $o(T)$. The regret incurred up to time-step $\tau$ is bounded as in the previous case, with an additional cost of at most 1 for the single mismatch round (since $|\Delta_i^\phi| \leq 1$). Thus total $\Phi_i$-regret is $O(m\sqrt{T}) + o(T)$.

Therefore, $\mathcal{L}_i$ is a *no-$\Phi_i$-regret* learner.

## C.2. Transforming CCE to a Symmetric CCE

In this section we discuss how any CCE $\mu$ can be transformed into a symmetric CCE without changing the sum of players' expected utilities.

For $p \in \mathcal{P}$, define $g(p) := (p - c) f(p)$, and for a tuple $x \in \mathcal{P}^n$ define

$$\min(x) := \min_{j \in [n]} x_j.$$

Fix any tuple $x \in \mathcal{P}^n$. Let $t(x) := \big|\{j : x_j = \min(x)\}\big|$. Now we have the following:

$$\sum_{i=1}^n u_i(x) \;=\; t(x) \cdot \frac{g(\min(x))}{t(x)} \;=\; g(\min(x)). \tag{1}$$

Let $S_n$ be the set of all permutations of $[n]$. For $\pi \in S_n$ and $x \in \mathcal{P}^n$, write $(\pi x)_i := x_{\pi(i)}$. We now construct a distribution $\bar{\mu}$ over $\mathcal{P}^n$ as follows: sample $x \sim \mu$ and independently sample a permutation $\pi$ uniformly from $S_n$, and output the permuted tuple $\pi x$.

For any player $i \in [n]$ and any fixed deviation $p \in \mathcal{P}$,

$$\mathbb{E}_{x \sim \bar{\mu}}[u_i(x)] = \frac{1}{n!} \sum_\pi \mathbb{E}_{x \sim \mu}[u_{\pi(i)}(x)] \;\geq\; \frac{1}{n!} \sum_\pi \mathbb{E}_{x \sim \mu}[u_{\pi(i)}(p, x_{-\pi(i)})] = \mathbb{E}_{x \sim \bar{\mu}}[u_i(p, x_{-i})],$$

Hence, $\bar{\mu}$ is also a CCE. Due to (1), we have the following:

$$\mathbb{E}_{x \sim \bar{\mu}}\left[\sum_i u_i(x)\right] = \frac{1}{n!} \sum_\pi \mathbb{E}_{x \sim \mu}\left[\sum_i u_{\pi(i)}(x)\right] = \frac{1}{n!} \sum_\pi \mathbb{E}_{x \sim \mu}[g(\min(x))] = \mathbb{E}_{x \sim \mu}\left[\sum_i u_i(x)\right]. \tag{2}$$

As $\bar{\mu}$ is permutation-invariant, all players have the same expected utility under $\bar{\mu}$. Hence, we have the following for each $i \in [n]$:

$$\mathbb{E}_{x \sim \bar{\mu}}\big[u_i(x)\big] = \frac{1}{n} \mathbb{E}_{x \sim \bar{\mu}}\Big[\sum_{j=1}^n u_j(x)\Big] = \frac{1}{n} \mathbb{E}_{x \sim \bar{\mu}}\big[g(\min(x))\big]. \tag{3}$$

Next, define the map $T : \mathcal{P}^n \to \mathcal{P}^n$ by
$$T(x) := (\min(x), \ldots, \min(x)).$$

Let $\nu$ be a distribution over $\mathcal{P}^n$ defined as follows: sample $x \sim \bar{\mu}$ and output $T(x)$. Then $\nu$ is supported on tuples with identical coordinate values. Also, since $\min(T(x)) = \min(x)$, we have $g(\min(T(x))) = g(\min(x))$, so

$$\mathbb{E}_{y \sim \nu} \Big[ \sum_i u_i(y) \Big] = \mathbb{E}_{x \sim \bar{\mu}} \big[ g(\min(T(x))) \big] = \mathbb{E}_{x \sim \bar{\mu}} \big[ g(\min(x)) \big]. \tag{4}$$

Now by combining (3) and (4) we get the following:

$$\mathbb{E}_{y \sim \nu}[u_i(y)] = \frac{1}{n} \mathbb{E}_{y \sim \nu} \Big[ \sum_j u_j(y) \Big] = \frac{1}{n} \mathbb{E}_{x \sim \bar{\mu}} \big[ g(\min(x)) \big] = \mathbb{E}_{x \sim \bar{\mu}}[u_i(x)]. \tag{5}$$

Fix a player $i \in [n]$ and a deviation price $q \in \mathcal{P}$ such that $q \geq c$. For any tuple $x$, recall that in the tuple $y = T(x)$, so all coordinate values in $y_{-i}$ is equal to $\min(x)$. Therefore, we have the following for all $x \in \mathcal{P}^n$:

$$u_i(q, y_{-i}) \;\leq\; u_i(q, x_{-i}) \tag{6}$$

The above follows due to following facts:

- If $q < \min(x)$, then $q$ is strictly below every opponent price in both $x_{-i}$ and $y_{-i}$.

- If $q > \min(x)$, then $u_i(q, y_{-i}) = 0$.

- If $q = \min(x)$, then $u_i(q, y_{-i}) = \frac{g(\min(x))}{n}$ whereas $u_i(q, x_{-i}) = \frac{g(\min(x))}{t+1}$ where $t := |\{j \neq i : x_j = m\}|$.

Now we have the following:

$$\mathbb{E}_{y \sim \nu} \big[ u_i(q, y_{-i}) \big] = \mathbb{E}_{x \sim \bar{\mu}} \big[ u_i(q, T(x)_{-i}) \big] \leq \mathbb{E}_{x \sim \bar{\mu}} \big[ u_i(q, x_{-i}) \big]. \tag{7}$$

Since $\bar{\mu}$ is a CCE, and using (5) and (7), we have the following:

$$\mathbb{E}_{y \sim \nu}[u_i(y)] = \mathbb{E}_{x \sim \bar{\mu}}[u_i(x)] \geq \mathbb{E}_{x \sim \bar{\mu}}[u_i(q, x_{-i})] \geq \mathbb{E}_{y \sim \nu}[u_i(q, y_{-i})]. \tag{8}$$

Since $i$ and $q$ were chosen arbitrarily, $\nu$ is a CCE.

Finally, we get the following by combining (2) and (4) to get

$$\mathbb{E}_{y \sim \nu} \Big[ \sum_i u_i(y) \Big] = \mathbb{E}_{x \sim \bar{\mu}} \Big[ \sum_i u_i(x) \Big] = \mathbb{E}_{x \sim \mu} \Big[ \sum_i u_i(x) \Big].$$

This completes the proof of our claim that any CCE can be transformed into a symmetric CCE without changing the sum of players' expected utilities.

