# OpenReview forum: "Revisiting the Bertrand Paradox via Equilibrium Analysis of No-regret Learners"
_ICML.cc/2026/Conference — ICML 2026 regular_

### Official Review · Reviewer_ihPm · 2026-03-11

**Soundness:** 4
**Presentation:** 4
**Significance:** 4
**Originality:** 3
**Overall Recommendation:** 6
**Confidence:** 5

**Summary:**

This paper studies equilibrium outcomes of no-regret learners in the discrete Bertrand pricing game with arbitrary non-increasing demand. It characterizes when high transaction prices can be sustained (CCE under no-external-regret) versus when prices are driven to marginal cost (CE under no-swap-regret), both in symmetric and asymmetric cost settings.

**Compliance With Llm Reviewing Policy:**

Affirmed.

**Key Questions For Authors:**

I will be very interested to learn about:
- The economic interpretation of the CCE vs CE price gap by the author?
- The tightness of the constant in Theorem 2.1: is this constant tight, or is it an artifact of the proof technique? Is there a matching lower bound showing that no CCE can achieve a constant factor better than that?

**Limitations:**

The n-player asymmetric analysis is absent and there is no discussion on its impossibility or anything related. But this is also a broad and complicated question that might require another paper to answer. I would appreciate it if the author discuss a bit about how hard the problem is and why they chose to omit it in the paper.

**Strengths And Weaknesses:**

## Strengths

The central question answered in this paper — what prices emerge when firms learn via no-regret algorithms —  is of high interest to both computer science and economic community. The paper cleanly separates the roles of external vs. swap regret and presents a satisfying conceptual dichotomy: weaker regret notions permit collusive-like outcomes; stronger ones enforce competitive pricing.

This paper qualifies as a technically solid paper. The proofs are non-trivial. Theorem 2.1 constructs an explicit symmetric CCE achieving a constant fraction (1/4e^2) of monopoly utility for *any* non-increasing demand. This is a very meaningful generalization beyond prior work. Theorem 2.5's exponential decay result is sharp and practically informative.

Assymetric cost analysis is very interesting and rare. The asymmetric duopoly results (Theorems 2.6–2.8) are genuinely surprising: a no-external-regret learner competing against a no-swap-regret learner can *still* sustain high prices for both players. This counterintuitive finding adds depth and extra interest.

The numerical experiments opens up new interesting directions. Figure 2 reveals that no-swap-regret dynamics can produce varied outcomes depending on learning rates. This is very interesting and motivates further theory.

## Weaknesses and Suggestions

The paper is very good in itself as a theory paper. I will make the following suggestions:

- Economic Interpretation of CCE vs. CE Gap. The paper would benefit from a short discussion of what the external/swap regret distinction means *economically* — e.g., swap regret corresponds to more sophisticated counterfactual reasoning by firms. This would broaden appeal to economics readers.
- Maybe say something about n-player asymmetric setting: results beyond duopoly in the asymmetric setting are absent. Even a brief impossibility argument or conjecture would round out the theory and makes the paper more complete.

---

> ### Author Rebuttal · Authors · 2026-03-29
>
> We thank the reviewer for the very positive assessment of the paper and for the thoughtful suggestions on how to strengthen the exposition. Below we respond to the main comments.
>
> ## Comment 1
> > The paper would benefit from a short discussion of the economic interpretation of the CCE versus CE gap, for example that swap regret corresponds to more sophisticated counterfactual reasoning by firms.
>
> ## Response
> Thank you for this helpful suggestion. We agree that this interpretation would broaden the paper’s appeal, especially to economics readers.
>
> Economically, coarse correlated equilibrium corresponds to the following guarantee: after observing the realized sequence of play, a firm cannot improve by replacing its entire strategy with one fixed price used in every round. This captures a relatively limited form of retrospective reasoning. In contrast, correlated equilibrium corresponds to a stronger guarantee: after observing the sequence of prices it actually played, a firm cannot improve by systematically replacing some chosen prices with alternative prices. This is a more refined form of counterfactual reasoning, since the firm is allowed to condition its deviations on its own action.
>
> ## Comment 2
> > Results beyond duopoly in the asymmetric setting are absent. Even a brief impossibility argument or conjecture would help explain this omission.
>
> ## Response
> The reason we do not state an $n$-player asymmetric analogue of our symmetric result is that such a statement is trivially false in general. In asymmetric markets, it is possible for a low-cost player to capture a large utility while all higher-cost players receive zero utility, even as the number of players grows. For example, if one player has marginal cost $0$ and all other players have marginal cost $1$, then the profile in which the low-cost player posts price $1 - 1/k$ and all other players post $1$ is a CCE. In that outcome, the low-cost player obtains utility $1 - 1/k$, which does not vanish with the number of players. Thus, unlike in the symmetric setting, there is no general exponential decay phenomenon in the asymmetric case.
>
> ## Comment 3
> > Is the constant in Theorem 2.1 tight, or is it an artifact of the proof technique?
>
> ## Response
> We believe the constant in Theorem 2.1 is an artifact of the proof technique and is not tight. In particular, based on our numerical experiments, we suspect that the correct constant should be $1/e$.

---

> > ### Author Rebuttal · Reviewer_ihPm · 2026-04-03
> >
> > I thank and appreciate the authors for their detailed response which answered my question. I maintain my assessment.

---

### Official Review · Reviewer_4caq · 2026-03-11

**Soundness:** 3
**Presentation:** 3
**Significance:** 3
**Originality:** 3
**Overall Recommendation:** 4
**Confidence:** 4

**Summary:**

This paper studies the equilibrium outcomes reached by no-regret learning algorithms in repeated Bertrand pricing games with general non-increasing demand functions.  Firms have potentially different fixed production costs $c_1 \ge c_2 \ge ... \ge c_n$ and they use no-regret algorithms to choose prices $x_i$ from a discretized set; the firm choosing the lowest price obtains utility $(x_i - c_i) f(x_i)$ with a general demand function $f(x_i)$.

For the case of symmetric cost $c_i = c$, the authors show 5 theoretical results:

(1) There exists a coarse correlated equilibrium (CCE), which is the outcome reached by some no-external-regret algorithms, that sustains supra-competitive (higher than $c$) prices such that the players obtian a high positive utility (instead of 0 utility as in the competitive equilibrium).  (Theorem 2.1)

(2) At any correlated equilibrium (CE), which is reached by no-swap-regret algorithms, the prices must be close to c, hence the players obtain nearly 0 utility.  (Theorem 2.2)

(3) If one player uses no-external-regret algorithm and another player uses no-swap-regret algorithms, there exist cases that sustain high prices, where the two players obtain positive utility.  (Theorem 2.3)

(4) But as long as two or more players use no-swap-regret algorithms, the price will be down to the cost.  (Theorem 2.4).

(5) The highest utility that no-external-regret algorithms (CCE) can obtain decreases exponentially as the number of firms increases.

The theoretical results for the asymmetric-cost case are similar to results (1) (2) (3) for the symmetric case.

Experiments are provided to explore:

* the ratio between the player's utility under CCE and the monopoly utility
* and the distribution of prices chosen by no-swap-regret algorithms in the long run.

**Compliance With Llm Reviewing Policy:**

Affirmed.

**Final Justification:**

Same as my "Rebuttal Acknowledgement".

**Key Questions For Authors:**

(Q1) Although you showed in Theorem 2.5 that the utilities of players in any CCE decrease exponentially with the number of players, does that mean the actual prices they choose converge to the cost c (so the consumers get the best price)?

**Limitations:**

### Suggestion

* There is a typo in line 99: $(x_i - c_i), f(x_i)$
* Page 6, the sentence "For asymmetric-cost setting, Jann & Schottmüller (2015) previously observed in the continuous Bertrand pricing game that, for an arbitrary non-increasing demand function, the second player which has the higher marginal cost, its expected utility is zero." has a grammatical error.

**Strengths And Weaknesses:**

_Soundness:_ I went over all proof sketches and read the proofs of Theorems 2.1 and 2.2 in detail.  They look sound to me. The experiments part also looks good.

_Presentation_:

(+) The paper is clearly written and well structured.  My only mild suggestion is to separate the theoretical results for symmetric cost and asymmetric cost settings (currently in two subsections) into two sections.

(+) The work properly positions itself in the context of prior literature. The authors acknowledge that prior works Nadav & Piliouras (2010) and Bichler et al (2024) have observed supra-competitive CCE in Bertrand pricing games with linear demand functions. Then, this work generalizes the conclusion to arbitrary non-increasing demand functions.  Moreover, Remark 1 explicitly says that this doesn't hold for all no-external-regret algorithms and mentions some works that analyze specific algorithms that do not sustain supra-competitive prices.  This clearly defines the contribution of this work to the literature.



_Originality & Significance_ :

(++) Among the many conclusions of this work, I think the most important one is the existence of supra-competitive CCE for any general non-increasing demand function (Theorem 2.1).  Prior works Nadav & Piliouras (2010) and Bichler et al (2024) observed similar conclusions but for the special linear demand function, and this work generalizes to any non-increasing demand function.

(-) Despite the generality for any non-increasing demand function, the above result only says that _some_ CCEs sustain supra-competitive prices, not all CCEs, so it's possible that only some "unnatural" no-external-regret algorithms can converge to those supra-competitive CCEs.  In fact, prior works have shown that the CCE reached by the popular class of mean-based algorithms (which includes Hedge) must be competitive (Deng et al (2022), Bichler et al (2024)).  It would be nice if the authors could provide some evidence that natural no-external-regret algorithms can converge to supra-competitive CCE.

(++) The second most important conclusion, I think, is that supra-competitive prices can occur even when one player runs a no-swap-regret algorithm, with another player running a no-external-regret algorithm (Theorem 2.3).  Such an "asymmetric regret/algorithms" perspective might inspire future research on other types of games to explore whether such an asymmetric equilibrium notion deviates or coincides with the standard CE, CCE, or Nash equilibrium notions.

(+) The result that any CE must be competitive also generally holds for any non-increasing demand function.  The authors acknowledge that the same result for the continuous Bertrand pricing game is known, and their contribution lies in proving the result for the discretized case, which is practically relevant.  I think this new result should be published somewhere to be known by the community.

(+/-) The conclusion that the players' utilities at any CCE decrease exponentially with the number of players is interesting.  It suggests that collusion is no longer an issue if the number of firms is large enough and they run simple algorithms like no-external-regret algorithms.  But I need to verify an issue with the authors before further evaluation; see (Q1).





------

Deng et al (2022).  Nash convergence of mean-based learning algorithms in first price auctions. The ACM Web Conference.

Bichler et al (2024). Online optimization algorithms in repeated price competition: Equilibrium learning and algorithmic collusion. arXiv.

---

> ### Author Rebuttal · Authors · 2026-03-29
>
> We thank the reviewer for the positive assessment of the paper. Below we address the main questions and comments.
>
> ## Comment 1
> > The paper shows that some CCEs sustain supra-competitive prices, but perhaps only for "unnatural" no-external-regret algorithms. It would be nice to provide evidence that natural no-external-regret algorithms can converge to such CCEs.
>
> ## Response
> We agree that this is an important question, although the term "natural" is itself difficult to define precisely. Our goal in this paper is not to claim that common algorithm classes necessarily converge to supra-competitive CCEs. Rather, our contribution is to show that the no-external-regret property alone is not sufficient to rule out such outcomes. In other words, the result identifies the limitation of the regret notion itself, independent of any specific algorithmic implementation. We believe this motivates future work on formalizing what constitutes a natural learning rule in this setting and on identifying which algorithm classes do or do not permit supra-competitive outcomes.
>
> ## Question 1
> > In Theorem 2.5, utilities in any CCE decrease exponentially with the number of players. Does that also imply that actual transaction prices converge to the cost $c$?
>
> ## Response
> Yes. In our setting, the total utility is directly proportional to the gap between transaction prices and the marginal cost $c$. Therefore, if the total utility approaches zero as the number of players grows, the transaction prices must also approach $c$.

---

> > ### Author Rebuttal · Reviewer_4caq · 2026-04-03
> >
> > My quesion is resolved. I keep a positive assessment.  My last (+/-) comment is now a (+).
> >
> > I'd suggest the authors adding "transacting prices approaches $c$" to Theorem 2.5 because it is the prices (not the firms' utilities) that determine whether the CCE is supra-competitive and determine the consumer welfare.

---

### Official Review · Reviewer_hC5B · 2026-03-12

**Soundness:** 3
**Presentation:** 4
**Significance:** 3
**Originality:** 3
**Overall Recommendation:** 5
**Confidence:** 4

**Summary:**

The present paper considers the question of potential outcomes and dynamics of no-phi-regret learners in the standard complete information Bertrand competition. More generally, they consider the problem of what prices can be sustained in equilibrium outcomes that can be attained by no-regret learners. They consider arbitrary valid demand functions, symmetric and asymmetric costs, and symmetric and asymmetric general no-phi-regret learners. The paper provides several novel theoretical insights and backs it up with some empirical experiments.

**Compliance With Llm Reviewing Policy:**

Affirmed.

**Final Justification:**

The paper is a solid contribution to the conference. I clearly vote for acceptance.

**Key Questions For Authors:**

1. Your construction of no-$\Phi$ regret learners that converge to a specific $\Phi$ equilibrium necessitates a certain form of coordination (by already knowing of the specific equilibrium beforehand). Did the authors explore whether one can exclude certain equilibria when this kind of coordination among learners is not allowed?

2. Please comment on my statements about the broader impact of your work.

**Limitations:**

yes

**Strengths And Weaknesses:**

In general, I do not have too much to say. I like the paper and the results and vote for acceptance even though I estimate the overall impact to be small (see below).

## Main strengths

The considered settings and extensions to previous literature are clear. The extensions to general demand functions and asymmetric cases are substantial. I particularly like that the authors demonstrate the limitations of no-external regret as single property to characterize observed outcomes in one of the most fundamental economic interactions. While this insight itself is not new, the simplicity and importance of the Bertrand competition is a very compelling example to make this point clear. At the same time they establish that while one no-swap regret learner is insufficient to ensure competitive prices, two already suffice.

The statements, settings, and assumptions are explained clearly and presented in an easy-to-understand way. The mathematical proofs are clearly stated. However, I did not check them all in detail. Overall, the writing is easy to follow and the paper extends previous work.

## Main weaknesses

The main weakness that I see is the overall impact of the paper. The main takeaways from this work were already made before. In particular, when considering the question of the Betrand paradox. These include (some of these overlap or imply one another): (i) there exist CCEs that include supracompetitive prices in several models of the Bertrand competition, (ii) these CCE might be learned by no-external regret learners, (iii) no-external regret is insufficient to ensure convergence to competitive prices or ensure convergence to supracompatitive ones, (iv) a single no-internal regret learner is also insufficient to enable full predictability. Other works instead establish convergence by other learners such as mean-based ones. Therefore, this paper is providing little new insight for the bigger question of dynamics under no-regret in economically motivated interactions.

## Minor comments
- A technical comment that I am sure that the authors are aware of. The authors state: “It is well known that when all players incur low external regret, the empirical joint distribution of prices converges to a coarse correlated equilibrium.” A similar statement is made for CEs. The standard result does not ensure convergence to a CCE. Instead the empirical distribution eventually satisfies the CCE constraints. That is, one can only speak of convergence to the set of CCE, not to a specific one.
- Typo: page 2, should be a \cdot instead of a comma in the utility function.
- I have some small wording issues that overemphasize some things unnecessarily:
1. The results by Nadav & Piliouras (2010) are arguably not very “recent” anymore.
2. The authors state: “can we guarantee that player 2 receives zero expected utility under any such Φ-correlated equilibrium? Surprisingly, the answer is no, as we show in the following theorem.” – Why is that surprising? It strengthens the statement from before.

---

> ### Author Rebuttal · Authors · 2026-03-29
>
> We thank the reviewer for the positive and detailed evaluation of our work. Below we respond to the main comments and questions.
>
> ## Question 1
> > Your construction of no-regret learners that converge to a specific equilibrium requires a form of coordination, since the target equilibrium is effectively built into the construction. Did the authors explore whether certain equilibria can be excluded when this kind of coordination is not allowed?
>
> ## Response
> We agree this is a fundamental and very interesting open question. Our current results are existential: they show what can occur under general no-regret guarantees, but they do not attempt to characterize which equilibria remain reachable under additional restrictions on coordination, initialization, or algorithm design. Some prior work has shown that particular classes of algorithms, such as mean-based no-regret learners, converge to much smaller subsets of coarse correlated equilibria. However, a general theory identifying which outcomes are excluded for broader classes of no-regret learners is still largely absent from the literature. We view this as an important direction for future work, and one of the motivations for the present paper is precisely to highlight this gap.
>
> ## Comment 2
> > The main weakness is the overall impact of the paper. Several of the broad takeaways were already known from prior work on Bertrand competition and no-regret learning. Please comment on the broader impact of your work.
>
> ## Response
> While prior work had established the CCE result in the symmetric setting for linear demand functions, the corresponding question remained open for general non-increasing demand functions and for the asymmetric setting, including the linear-demand case. This is one of the gaps our paper resolves. In addition, we show that even if one of the two no-external-regret learners also satisfies the no-swap-regret property, high transaction prices may still persist. To our knowledge, this is a new insight that has not been established in prior works. We further show that sustaining high transaction prices becomes more difficult as the number of players increases, which may partly explain why real-world markets rarely feature a very large number of competitors. Finally, our numerical experiments reveal that no-swap-regret dynamics can generate a range of outcomes depending on the learning rates. We hope these findings will motivate further theoretical work on dynamics under no-regret learning beyond the standard focus on convergence rates, and ultimately broaden the impact of this line of research at the intersection of economics and learning theory.
>
> ## Comment 3
> > The authors state: “can we guarantee that player 2 receives zero expected utility under any such Φ-correlated equilibrium? Surprisingly, the answer is no, as we show in the following theorem.” – Why is that surprising?
>
> ## Response
> We called this result surprising because the asymmetric-cost setting creates an additional source of strategic asymmetry that could plausibly make competitive outcomes easier to enforce. In particular, one might expect that a no-swap-regret player with a lower marginal cost could profitably undercut the other player by setting a price below the latter’s marginal cost, thereby preventing the higher-cost player from sustaining elevated prices. Theorem 2.8 shows that this intuition is false: even in this asymmetric setting, one no-swap-regret player is still insufficient to force the other player’s expected utility to zero.

---

> > ### Author Rebuttal · Reviewer_hC5B · 2026-04-02
> >
> > The authors addressed my concerns adequately. I acknowledge their non-trivial contribution to the field. However, I am still a little sceptical of the overall impact that work will have.
> > Anyway, I think this work is a nice addition to the conference. As my score has been high already, I will keep it as it is.

---

### Official Review · Reviewer_d8mq · 2026-03-12

**Soundness:** 2
**Presentation:** 4
**Significance:** 3
**Originality:** 3
**Overall Recommendation:** 4
**Confidence:** 4

**Summary:**

This paper studies questions regarding no-regret learning algorithms in discrete-price-level Bertrand competition with non-increasing demand functions. The key feature of the model is that demand can differ across price levels; therefore, one player can have different utility when capturing the whole market at different price levels.

The paper first considers the case where the players have symmetric marginal costs. It shows that there exists a coarse correlated equilibrium  (CCE) where the expected utility of the players is a constant fraction of that attained at the highest price. The paper then shows that in any correlated equilibrium (CE), the players' expected utility lies between 0 and the marginal cost, which vanishes as the discretization becomes finer. However, a player with a guarantee of non-swap beneficial deviation corresponding to CE can not unilaterally ensure non-cooperation with no external regret player to sustain high utility. The paper additionally shows that the increasing competition reduces the fraction of the monopoly utility that non-external regret learners can achieve.

The paper then considers the cases when the players have asymmetric marginal costs and obtains similar results.

Finally, the paper conducts numerical experiments to supplement its theoretical findings, motivating the need to identify more fine-grained features of no-swap-regret learners that affect the convergence.

**Compliance With Llm Reviewing Policy:**

Affirmed.

**Final Justification:**

The paper made a solid contribution to the study of auctions with no-regret learners. The rebuttal addressed most of my main concerns.

**Key Questions For Authors:**

1. In the model, all prices and demand are normalized to 1. I would like to know if the constants in Theorem 2.1 ($1/4e^2$), Theorem 2.3 ($\lambda_0$) are scale-invariant? If the constant can change with the magnitude of the highest price level, it would weaken the result to some extent.

2. The results papers presented relate players' (sellers) utility to the monopoly utility. On the other hand, the paper interprets these results as the ability to "sustain high transaction prices." Since the model features an arbitrary demand function, is it possible that high seller utility does not necessarily imply high transaction prices?

3. Could the author say more about the relationship between Theorem 2.3 and the example from Section 5 of Hartline et al., 2025, "Regulation of Algorithmic Collusion, Refined: Testing Pessimistic Calibrated Regret"? The latter also considers the setting where one swap-deviation-free player can not unilaterally ensure non-cooperation with another constant-deviation-free player to avoid a high payoff. I'm not sure if the latter is/or can be reformulated as a Bertrand pricing game. But is Theorem 2.3 a generalization of the latter?

# Minor Question / Comments for Authors
The qualitative nature of the result, Theorem 2.8 in the asymmetric costs setting, is the same as Theorem 2.3 in the symmetric costs setting. I do not find that "Surprisingly, the answer is no", just feel unsure when I got that part of the paper. Should the authors have better intuition, please elaborate.

**Limitations:**

Yes. For example, the remark under Theorem 2.1 clearly states the scope of the claim; the paper does not claim that the bad CCE can be reached by some existing learning dynamics.

**Strengths And Weaknesses:**

# Strength
The paper presents theoretical results from a standard model that are clean, easily understandable, and readily comparable to the existing literature. I also find the presentation very good and easy to understand.

# Weakness (also refer to the questions for more details)
1. The magnitude of constants that capture the multiplicative factor to monopoly utility is very small (e.g., $1/4e^2 \approx $ 3%) despite being an absolute constant independent of the discretization.
2. There are some gaps between the interpretation of the results given by the paper and what the theorem states.

---

> ### Author Rebuttal · Authors · 2026-03-29
>
> We thank the reviewer for the positive assessment of the paper. Below we respond to each question and comment in turn.
>
> ## Question 1
> > In the model, all prices and demand are normalized to 1. Are the constants in Theorem 2.1 and Theorem 2.3 scale-invariant? If the constant changes with the magnitude of the highest price level, that would weaken the result.
>
> ## Response
> Yes, the constants in Theorems 2.1 and 2.3 are scale-invariant. The reason is that these constants are defined as ratios between the utility achieved in the duopoly setting and the monopoly utility. If the price scale is multiplied by a common factor, then both the numerator and denominator scale proportionally, so the ratio remains unchanged.
>
> ## Question 2
> > The paper relates seller utility to monopoly utility, but interprets the results as the ability to "sustain high transaction prices." Since the model allows arbitrary demand functions, is it possible that high seller utility does not necessarily imply high transaction prices?
>
> ## Response
> In our model, high seller utility does imply high transaction prices. Intuitively, utility is bounded above by the transaction price, since demand is normalized between 0 and 1 and costs are nonnegative. Therefore, if expected utility is high, the transaction prices must also be high on average.
>
> ## Question 3
> > Could the authors say more about the relationship between Theorem 2.3 and the example from Section 5 of Hartline et al. (2025)? Is Theorem 2.3 a generalization of that result?
>
> ## Response
> We do not view Theorem 2.3 as a direct generalization of the result in Section 5 of Hartline et al. (2025). As we understand it, that section considers a setting in which both players satisfy only no-external-regret guarantee and one of the player uses mean-based no-regret learner. By contrast, our theorem studies a mixed-regret setting, where one player satisfies a no-swap-regret guarantee and the other only satisfies no-external regret. In addition, the focus of their example is to show that prices can remain above those of correlated equilibrium, whereas our result shows something stronger in the Bertrand setting: even with one no-swap-regret player, the outcome can still sustain a constant fraction of monopoly utility.
>
> ## Comment 4
> > The qualitative result in the asymmetric-cost setting (Theorem 2.8) is similar to that in the symmetric-cost setting (Theorem 2.3). Why is the statement "Surprisingly, the answer is no" appropriate here?
>
> ## Response
> We called this result surprising because the asymmetric-cost setting creates an additional source of strategic asymmetry that could plausibly make competitive outcomes easier to enforce. In particular, one might expect that a no-swap-regret player with a lower marginal cost could profitably undercut the other player by setting a price below the latter’s marginal cost, thereby preventing the higher-cost player from sustaining elevated prices. Theorem 2.8 shows that this intuition is false: even in this asymmetric setting, one no-swap-regret player is still insufficient to force the other player’s expected utility to zero.

---

> > ### Author Rebuttal · Reviewer_d8mq · 2026-04-01
> >
> > Thanks to the authors for the detailed responses to all my questions. I am pretty satisfied with the answers for Q2-Q4. I do still feel a little nervous about the fact that the constant multiplicative factor of monopoly revenue is very small, e.g., ~3%. Therefore, I will keep the current score for the paper I give. Since all other reviewers are also quite positive about the paper, this should not matter that much for the outcome.

---

### Decision · Program_Chairs · 2026-04-30

**Decision:**

Accept (regular)

**Comment:**

The paper studies equilibria reached by no-regret learners in a classical pricing game, proving theoretical results for different notions of equilibrium.  All reviewers like the paper, identifying strengths such as the meaningful model and results, the clear relation to the existing literature, and the nice conceptual messages.  There were minor concerns, most of which were resolved after the author response phase.  Overall I believe the paper would be a solid contribution to ICML.